# Quorum-sensing *agr* system of *Staphylococcus aureus* primes gene expression for protection from lethal oxidative stress

Magdalena Podkowik[1,2], Andrew I Perault[2,3], Gregory Putzel[2,3,4], Andrew Pountain[5], Jisun Kim[6], Ashley L DuMont[1], Erin E Zwack[3], Robert J Ulrich[1], Theodora K Karagounis[2,7], Chunyi Zhou[1,2], Andreas F Haag[8], Julia Shenderovich[2,3], Gregory A Wasserman[9], Junbeom Kwon[1], John Chen[10], Anthony R Richardson[11], Jeffrey N Weiser[3], Carla R Nowosad[12], Desmond S Lun[13], Dane Parker[6], Alejandro Pironti[2,3,4], Xilin Zhao[14], Karl Drlica[15,16], Itai Yanai[5,17], Victor J Torres[2,3], Bo Shopsin[1,2,3]*

[1]Department of Medicine, Division of Infectious Diseases, NYU Grossman School of Medicine, New York, United States; [2]Antimicrobial-Resistant Pathogens Program, New York University School of Medicine, New York, United States; [3]Department of Microbiology, NYU Grossman School of Medicine, New York, United States; [4]Microbial Computational Genomic Core Lab, NYU Grossman School of Medicine, New York, United States; [5]Institute for Systems Genetics; NYU Grossman School of Medicine, New York, United States; [6]Department of Pathology, Immunology and Laboratory Medicine, Center for Immunity and Inflammation, Rutgers New Jersey Medical School, Newark, United States; [7]Ronald O. Perelman Department of Dermatology; NYU Grossman School of Medicine, New York, United States; [8]School of Medicine, University of St Andrews, St Andrews, United Kingdom; [9]Department of Surgery, Northwell Health Lenox Hill Hospital, New York, United States; [10]Department of Microbiology and Immunology, Yong Loo Lin School of Medicine, National University of Singapore, Singapore, Singapore; [11]Department of Microbiology and Molecular Genetics, University of Pittsburgh, Pittsburgh, United States; [12]Department of Pathology, NYU Grossman School of Medicine, New York, United States; [13]Center for Computational and Integrative Biology and Department of Computer Science, Rutgers University, Camden, United States; [14]State Key Laboratory of Molecular Vaccinology and Molecular Diagnostics, School of Public Health, Xiamen University, Xiamen, China; [15]Public Health Research Institute, New Jersey Medical School, Rutgers University, New Yprk, United States; [16]Department of Microbiology, Biochemistry & Molecular Genetics, New Jersey Medical School, Rutgers University, Newark, United States; [17]Department of Biochemistry and Molecular Pharmacology, NYU Grossman School of Medicine, New York, United States

*For correspondence:
Bo.Shopsin@nyulangone.org

**Abstract** The *agr* quorum-sensing system links *Staphylococcus aureus* metabolism to virulence, in part by increasing bacterial survival during exposure to lethal concentrations of $H_2O_2$, a crucial host defense against *S. aureus*. We now report that protection by *agr* surprisingly extends beyond post-exponential growth to the exit from stationary phase when the *agr* system is no longer

turned on. Thus, *agr* can be considered a constitutive protective factor. Deletion of *agr* resulted in decreased ATP levels and growth, despite increased rates of respiration or fermentation at appropriate oxygen tensions, suggesting that Δ*agr* cells undergo a shift towards a hyperactive metabolic state in response to diminished metabolic efficiency. As expected from increased respiratory gene expression, reactive oxygen species (ROS) accumulated more in the *agr* mutant than in wild-type cells, thereby explaining elevated susceptibility of Δ*agr* strains to lethal $H_2O_2$ doses. Increased survival of wild-type *agr* cells during $H_2O_2$ exposure required *sodA*, which detoxifies superoxide. Additionally, pretreatment of *S. aureus* with respiration-reducing menadione protected Δ*agr* cells from killing by $H_2O_2$. Thus, genetic deletion and pharmacologic experiments indicate that *agr* helps control endogenous ROS, thereby providing resilience against exogenous ROS. The long-lived 'memory' of *agr*-mediated protection, which is uncoupled from *agr* activation kinetics, increased hematogenous dissemination to certain tissues during sepsis in ROS-producing, wild-type mice but not ROS-deficient (*Cybb*$^{-/-}$) mice. These results demonstrate the importance of protection that anticipates impending ROS-mediated immune attack. The ubiquity of quorum sensing suggests that it protects many bacterial species from oxidative damage.

## eLife assessment

This **important** study outlines how the agr quorum sensing system in *Staphylococcus aureus* confers long-lived protection against oxidative stress, thereby linking bacterial metabolism to virulence in this pathogen. While the findings, which are supported by **solid** data, seem at first glance to contradict earlier findings that show increased fitness of agr mutants under oxidative stress, the core conclusions of the study are well-substantiated. The topic of the paper holds broad relevance to microbiologists, especially those focusing on host-pathogen interactions and bacterial responses to ROS.

## Introduction

Innate, bactericidal immune defenses and antimicrobials act, at least in part, by stimulating the accumulation of ROS in bacteria (*Spaan et al., 2013*; *Drlica and Zhao, 2021*). Thus, understanding how *Staphylococcus aureus* and other bacterial pathogens manage ROS-mediated stress has important implications for controlling infections.

Knowledge of factors that govern the biology of ROS has advanced considerably in recent years. For example, studies have centered on how specific metabolic features, such as aerobic respiration, affect killing by ROS (*Kohanski et al., 2007*; *Lobritz et al., 2015*), and small-molecule enhancers of ROS-mediated lethality are emerging (*Shatalin et al., 2021*; *Shee et al., 2022*). Less well characterized is how defense against ROS and metabolism changes integrate with the virulence regulatory network that promotes *S. aureus* pathogenesis. The *agr* quorum-sensing system provides a way to study this dynamic: *agr* is a major virulence regulator that responds to oxidative stress ($H_2O_2$). The response occurs through a redox sensor in AgrA that attenuates *agr* activity, thereby increasing the expression of glutathione peroxidase (BsaA), an enzyme that detoxifies ROS (*Sun et al., 2012*). Whether protection from ROS also occurs from positive *agr* action is unknown and likely to be an important issue in the development of Agr-targeted therapies (*Khan et al., 2015*).

In cultured *S. aureus*, *agr* governs the expression of ~200 genes. Its two-part regulatory role is characterized by (1) increased post-exponential-phase production of toxins and exoenzymes that facilitate dissemination of bacteria via tissue invasion, and (2) decreased production of cell surface and other proteins that facilitate adherence, attachment, biofilm production, and evasion of host defenses (*Novick, 2003*; *Novick and Geisinger, 2008*). Thus, *agr* coordinates a switch from an adherent state to an invasive state at elevated bacterial population density. The invasive state would be facilitated by protection from host defense.

The *agr* locus consists of two divergent transcription units driven by promoters P2 and P3 (*Novick et al., 1995*). The P2 operon encodes the quorum-signaling module, which contains four genes, *agrB*, *agrD*, *agrC*, and *agrA*. AgrC is a receptor histidine kinase, and AgrA is a DNA-binding response regulator. AgrD is an autoinducing, secreted peptide derived from a pro-peptide processed by AgrB. The autoinducing peptide binds to and causes autophosphorylation of the AgrC histidine kinase, which

phosphorylates and activates the DNA-binding AgrA response regulator. AgrA then stimulates transcription from the P2 (RNAII) and P3 (RNAIII) promoters. RNAIII is a regulatory RNA that additionally contains the gene for delta-hemolysin (*hld*). The DNA-binding domain of AgrA contains an intramolecular disulfide switch (*Sun et al., 2012*). Oxidation leads to dissociation of AgrA from DNA, thereby preventing an AgrA-mediated down-regulation of the BsaA peroxidase.

When we used antimicrobials to study bacterial responses to lethal stress involving the accumulation of ROS, we found that inactivation (deletion) of *agr* reduces lethality arising from treatment with antimicrobials, such as fluoroquinolones, in a largely *bsaA*-dependent manner (*Kumar et al., 2017*). Thus, oxidation sensing appears to be an intrinsic checkpoint that ameliorates the endogenous oxidative burden generated by certain antimicrobials. Surprisingly, deletion of *agr increases* the lethal effects of exogenous $H_2O_2$ (*Kumar et al., 2017*), in contrast to the expected expression of the protective *bsaA* system (*Sun et al., 2012*). Thus, *agr* must help protect *S. aureus* from exogenous ROS, a principal host defense, through mechanisms other than *bsaA*.

In the present work we found that protection by wild-type *agr* against lethal concentrations of $H_2O_2$ was unexpectedly long-lived and (1) associated with decreased expression of respiration genes, and (2) potentially aided by defense systems that suppress the oxidative surge triggered by subsequent, high-level $H_2O_2$ exposure. The redox switch in AgrA, plus these additional protective properties, indicate that *agr* increases resilience to oxidative stress in *S. aureus* both when it is present and when it is absent. Thus, *agr* integrates protection from host defense into the regulation of staphylococcal virulence.

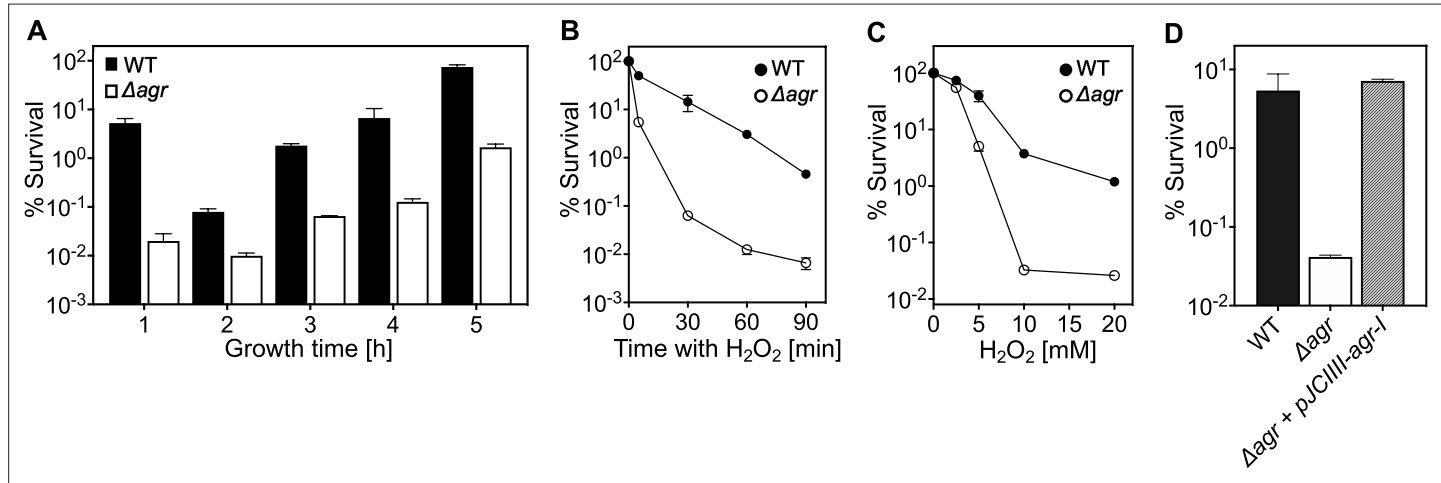

**Figure 1.** *agr* protects from killing by $H_2O_2$ throughout the growth cycle. (**A**) Effect of culture growth phase. Overnight cultures of *S. aureus* LAC wild-type (WT, BS819) or Δ*agr* (BS1348) were diluted ($OD_{600}$~0.05) into fresh TSB medium and grown with shaking from early exponential (1 h, $OD_{600}$~0.15) through late log (5 h, $OD_{600}$~4) phase. At the indicated times, early (undiluted) and late exponential phase cultures (diluted into fresh Tryptic Soy Broth (TSB) medium to $OD_{600}$~0.15) were treated with $H_2O_2$ (20 mM). After 60 min, aliquots were removed, serially diluted, and plated for determination of viable counts. Percent survival was calculated relative to a sample taken at the time of $H_2O_2$ addition. (**B**) Kinetics of killing by $H_2O_2$. Wild-type and Δ*agr* mutant strains were grown to early exponential ($OD_{600}$~0.15) and treated with 20 mM $H_2O_2$ for the times indicated, and percent survival was determined by plating. (**C**) Effect of $H_2O_2$ concentration on survival. Cultures prepared as in panel B were treated with the indicated peroxide concentrations for 60 min prior to plating and determination of percent survival. (**D**) Complementation of *agr* deletion mutation. Cultures of wild-type (WT) cells (BS819), Δ*agr* mutant (BS1348), and complemented Δ*agr* mutant carrying a chromosomally integrated wild-type operon (pJC1111-*agrI*) were treated with 20 mM $H_2O_2$ for 60 min followed by plating to determine percent survival. Data represent the means ± SD. from biological replicates (n=3).

The online version of this article includes the following figure supplement(s) for figure 1:

**Figure supplement 1.** Correlation of growth phase and *agr* expression.

**Figure supplement 2.** Correlation of lag-time and *agr*-mediated protection from $H_2O_2$-mediated killing.

**Figure supplement 3.** Extended lag phase and decreased growth rate and yield of an Δ*agr* mutant.

**Figure supplement 4.** Agr-mediated protection from $H_2O_2$-mediated killing among diverse *S. aureus* strains.

# Results

## *agr* protects *S. aureus* from lethal concentrations of H$_2$O$_2$ throughout the growth cycle

Because *agr* is a quorum-sensing regulon, maximal *agr* activity occurs during exponential growth (*Figure 1—figure supplement 1*) and is followed by a sharp drop during stationary phase (*Kumar et al., 2017*; *Geisinger et al., 2012*). Surprisingly, protection from H$_2$O$_2$ toxicity by wild-type *agr*, assessed by comparison with an *agr* deletion mutant, was observed throughout the growth cycle (*Figure 1A*). Indeed, maximal protection occurred shortly after overnight growth, long after induction and expression of *agr* transcripts. Comparison of survival rates of Δ*agr* mutant and wild-type cells, following dilution of overnight cultures and regrowth for 1 hr prior to challenge with 20 mM H$_2$O$_2$, revealed an initial rate of killing that was ~1000 fold faster for the Δ*agr* mutant (*Figure 1B*). Peroxide concentration dependence was observed up to 10 mM during a 60 min treatment; at that point, mutant survival was about 100-fold lower (*Figure 1C*). Complementation tests confirmed that the *agr* deletion elevated killing by H$_2$O$_2$ (*Figure 1D*).

We also monitored the time required for the wild-type *agr* survival advantage against H$_2$O$_2$ to manifest itself (*Figure 1—figure supplement 2*). Overnight cultures were not readily killed by H$_2$O$_2$, as expected from previous results with other lethal stressors (*Conlon et al., 2016*). Following dilution to fresh medium, wild-type survival dropped gradually, while mutant survival, although lower, was constant for 20 min. By 40 min, mutant survival exhibited a precipitous 10-fold drop not seen with wild-type cells (*Figure 1—figure supplement 2*). This drop in mutant survival correlated temporally with changes in cell density (*Figure 1—figure supplement 2*); i.e., the first cell division following dilution to fresh medium. Overall, the *agr*-mediated survival advantage during H$_2$O$_2$ exposure was absent in stationary-phase cells and small during lag phase (before exponential growth resumes), but it increased markedly during early growth.

Lag-time differences between strains were more obvious in experiments using less complex, chemically defined medium (CDM) with highly diluted starting cultures and automated growth analysis (*Figure 1—figure supplement 3*). In CDM, wild-type cells divided within ~150 min, while the lag times with the Δ*agr* mutant were more than 205 min (in Tryptic Soy Broth the lag time is 30 min for both). These observations suggest a novel *agr*-mediated decrease in time to enter exponential growth following dilution of stationary phase cultures. The poor killing of *agr* mutant cells by H$_2$O$_2$ early in lag phase is consistent with other work in which cells experiencing long lag times are less readily killed (*Fridman et al., 2014*), presumably due to remaining longer in a dormant, protected state. To focus on effects during growth, subsequent experiments were performed after incubation of overnight cultures for 1 hr in fresh Tryptic Soy Broth unless otherwise specified.

The elevating effect of *agr* inactivation on H$_2$O$_2$-mediated lethality was observed across a variety of *S. aureus* strains, although differences in wild-type survival were observed (*Figure 1—figure supplement 4*). Thus, *agr*-mediated protection from H$_2$O$_2$ appears to be common among *S. aureus* lineages.

## Expression of RNAIII and repression of Rot is required for protection from H$_2$O$_2$-mediated lethality

Δ*rnaIII* and Δ*agr* mutants showed identical loss of protection from H$_2$O$_2$-mediated killing (*Figure 2A*), indicating that protection is RNAIII-dependent. Since RNAIII represses translation of the downstream regulator *Rot* (*Geisinger et al., 2006*), a transcription factor having a key role in *agr* regulation of staphylococcal virulence, we also examined the effects of *rot* on the protective action of *agr* against H$_2$O$_2$. When the wild-type strain, a Δ*agr* mutant, a Δ*rot mutant,* and a Δ*agr* Δ*rot* double mutant were compared for survival following treatment with 20 mM H$_2$O$_2$, survival of the Δ*agr* Δ*rot* double mutant phenocopied that of the wild-type strain (*Figure 2B*): the *rot* deletion reversed the effect of an *agr* deficiency. These data are consistent with *agr* activity allowing induction of *rot*-repressed genes important for protection from peroxide (RNAIII repression of the Rot repressor).

When a low-copy-number plasmid expressing *rot* was introduced into a wild-type strain, the transformant was more readily killed by H$_2$O$_2$, indicating that the expression of *rot* is sufficient for increased lethality (*Figure 2C–D*). These data suggest that wild-type Rot down-regulates expression of protective genes. The observed epistatic effect of *agr* and *rot* did not apply to other downstream, potentially epistatic regulators, such as *saeRS*, *mgrA*, and *sigB* (*Figure 2—figure supplement 1*; *Bronesky*

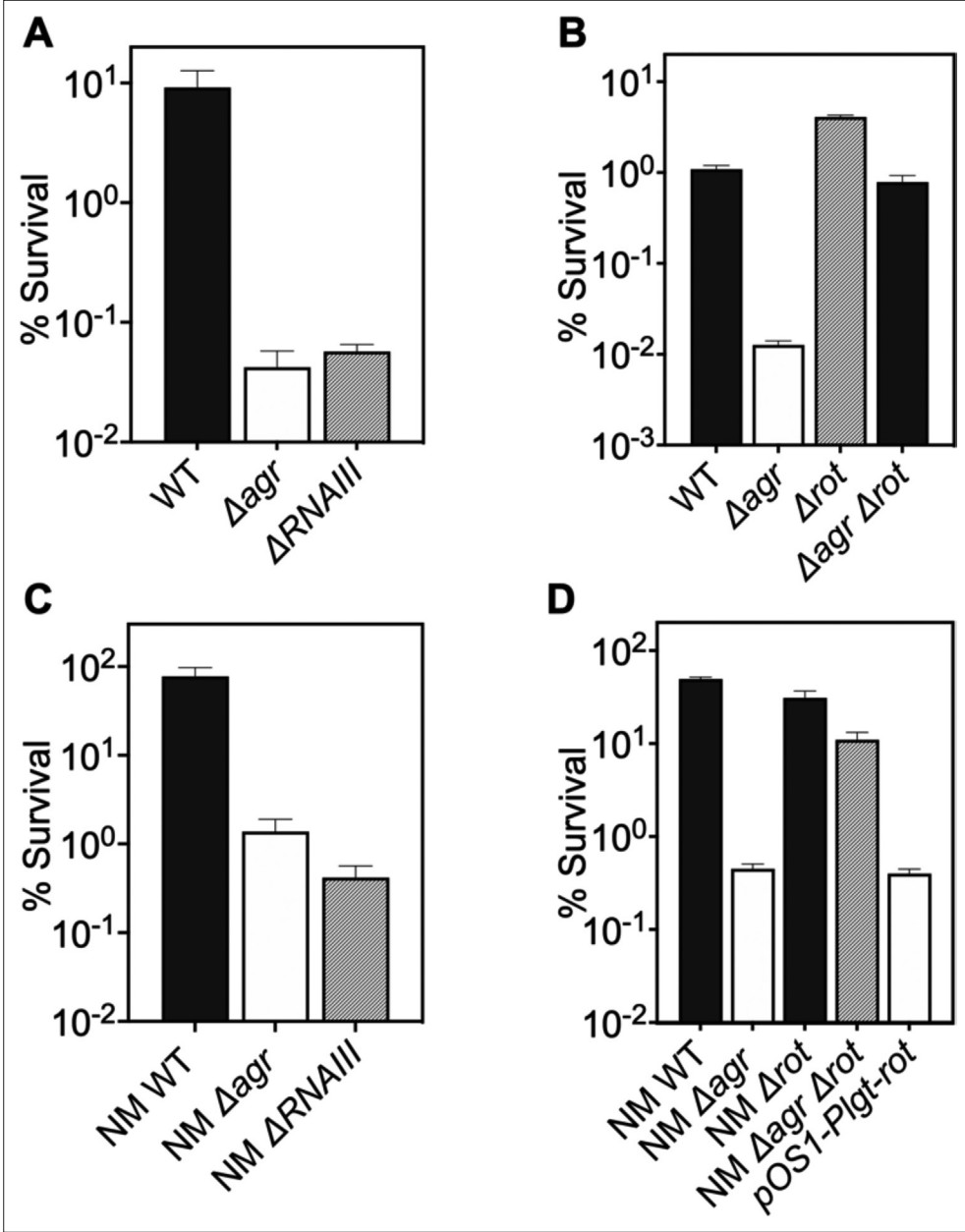

**Figure 2.** Involvement of *RNAIII* and *rot*-dependent pathways in *agr*-mediated protection from $H_2O_2$-mediated killing. Cultures were grown for 1 hr following dilution from overnight cultures to early log phase ($OD_{600}$~0.15) and then treated with 20 mM $H_2O_2$ for 60 min before determination of percent survival by plating and enumeration of colonies. (**A**) Wild-type LAC (WT, BS819), *Δagr* mutant (BS1348), and *ΔrnaIII* mutant (GAW183). (**B**) *Δrot* and *Δagr Δrot* double mutant (BS1302). (**C**) Wild-type (WT) strain Newman (NM, BS12), *Δagr* mutant (BS13), and *ΔRNAIII* mutant (BS669). (**D**) Overexpression of *rot*. Rot was expressed from a plasmid-borne wild-type *rot* (pOS1-*Plgt-rot*, strain VJT14.28). Data represent the mean ± SD. from biological replicates (n=3).

The online version of this article includes the following figure supplement(s) for figure 2:

**Figure supplement 1.** Deficiency of downstream global regulators does not differentially affect *agr*-mediated protection from $H_2O_2$-mediated cell death.

*et al., 2016*). Thus, the epistatic relationship between *agr* and protection from $H_2O_2$ appears to be *rot*-specific.

## *Agr*-mediated protection from $H_2O_2$ stress is kinetically uncoupled from *agr* activation

Since *agr*-mediated protection from $H_2O_2$ occurs throughout the growth cycle, it was possible that protection arises from constitutive, low-level *agr* expression rather than from autoinduction and thereby quorum sensing. To test for a requirement of quorum in the *agr*-mediated oxidative-stress phenotypes, we characterized the role of *agr* activation using a mixed culture strategy in which one strain, **an in-frame deletion mutant of *agrBD***, is activated in trans by AIP produced by a second, Δ*rnaIII* mutant strain (*Figure 3A*). The AIP-responsive Δ*agrBD* strain carried an intact RNAIII, while the Δ*rnaIII* mutant was wild-type for *agrBD*. As shown in *Figure 3B*, hemolytic activity (a marker for RNAIII) of the Δ*agrBD* mutant was restored by mixing it with the Δ*rnaIII* mutant strain that secreted AIP into the surrounding medium. This result confirmed that *agrCA*-directed *trans*-activation of RNAIII by AIP remained intact in the Δ*agrBD* mutant.

Mixed culture tests using these mutants, scored by differential plating for the presence of an erythromycin resistance marker in the Δ*agrBD* mutant, showed no protection from lethality of $H_2O_2$ when the two strains were mixed 1:1 immediately prior to growth from stationary phase (*Figure 3C*). Autoinducer accumulated during subsequent growth, activating *agr* expression and commencing protection from exogenous $H_2O_2$ (*Figure 3D–E*). During $H_2O_2$ treatment, the percentage of the Δ*agrBD* mutant (*rnaIII+*) increased while the percentage of the Δ*rnaIII* mutant decreased; this cis-acting result is consistent with the idea that pathways downstream from RNAIII, such as those regulated by *rot*, are the primary drivers of *agr*-mediated protection from $H_2O_2$. These results confirm an intimate link between *agr*-mediated protection and the quorum-controlled *agr* gene expression program of late exponential phase. However, after an overnight co-culture of the Δ*rnaIII* and Δ*agrBD* mutant strains, the Δ*agrBD* mutant demonstrated the same degree of protection expected for wild-type cells during exposure to $H_2O_2$ (*Figure 3F–H*). Thus, protection by *agr* after overnight co-culture extends to growth resumption from stationary phase, prior to reaching quorum, and therefore protection is uncoupled from the constraint of strict cell-density dependence. These results indicate that protection lasts long after maximal transcription of *agr*, when *agr* expression has largely halted (*Kumar et al., 2017*; *Geisinger et al., 2012*). This phenomenon is a critical feature of the *agr* system not appreciated in previous analyses of *agr* activation kinetics.

## agr deficiency increases transcription of genes involved in respiration and overflow metabolism in the absence of stress

To explore mechanisms underlying protection from $H_2O_2$, we performed RNA-seq with the Δ*agr* and wild-type strains after growth to late exponential growth phase, a point when *agr* expression is maximal. As expected, *agr* up-regulated the transcription of many known virulence genes (*Supplementary file 1*). The Δ*agr* strain showed elevated expression of genes involved in respiration (*cydA*, *qoxA-D*) and fermentation (*Fuchs et al., 2007*; *Pagels et al., 2010*), including *nrdGD*, alcohol dehydrogenases (*adhE* and *adh1*), and lactate dehydrogenases (*ldh*, *ddh*) (*Figure 4A* and *Supplementary file 1*). Increased respiration and fermentation are expected to increase energy generation. However, metabolic modeling of transcriptomic data showed a ~30% reduction in tricarboxylic acid (TCA) cycle and lactate flux per unit of glucose taken up by the Δ*agr* mutant (*Figure 4B*, *Supplementary file 1*). Additionally, intracellular ATP levels were ~50% lower in the Δ*agr* mutant compared to the wild-type control, suggesting reduced metabolic efficiency during exponential growth (*Figure 5A*). Moreover, although the *agr* deletion has little effect on growth in the rich medium in which RNA-seq was performed (*Somerville et al., 2002a*), analysis in nutrient-constrained medium (CDM) revealed decreased growth rate and yield of the Δ*agr* mutant relative to wild-type *S. aureus* (*Figure 1—figure supplement 3*). Collectively, these data suggest that Δ*agr* increases respiration and fermentation to compensate for low metabolic efficiency. Consistent with this idea, *agr* deficiency also increases ATP-yielding carbon 'overflow' pathways, as evidenced by increased acetate production (*Figure 5B*; *Sadykov et al., 2013*; *Somerville et al., 2002b*). The increase in accumulated acetate in the culture medium during exponential growth was largely consumed after 24 hr of growth (*Figure 5B*). Thus,

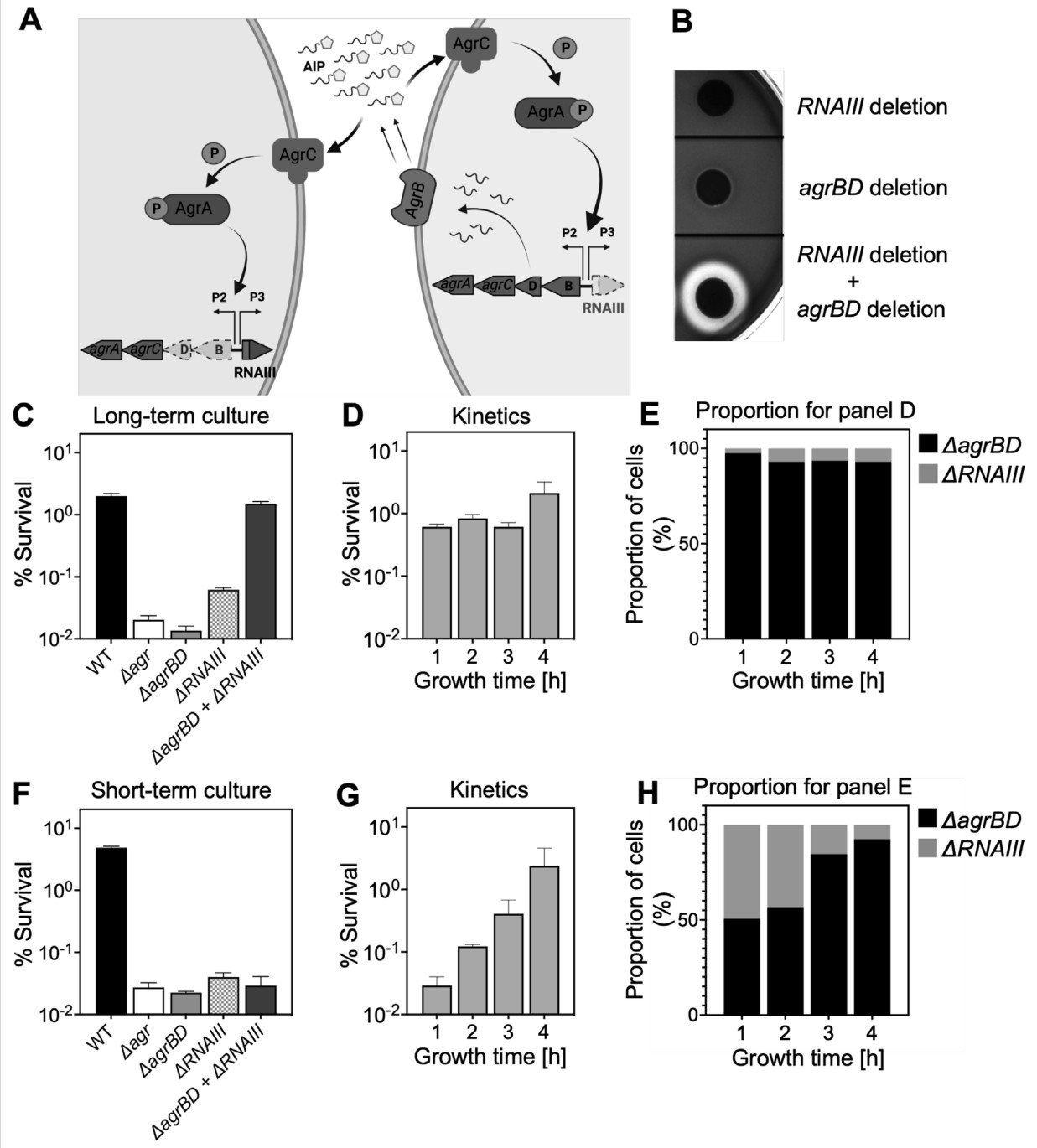

**Figure 3.** *Agr*-mediated protection from $H_2O_2$ stress is uncoupled from *agr* activation kinetics. (**A**) Assay design. An Δ*agrBD* **deletion mutant** (GAW130) was complemented in trans by the autoinducing product (AIP) of AgrBD in an Δ*rnaIII* (GAW183) mutant that produces AIP endogenously; AgrC activation in the Δ*agrBD* strain leads to downstream activation of RNAIII. *The agrBD* strain, engineered in-frame to avoid polar effects on downstream genes *agrC* and *agrA*, senses but does not produce an autoinducer. The Δ*rnaIII* mutant, constructed by replacement of *rnaIII* with a cadmium resistance cassette (*rnaIII::cadA*), produces autoinducer but lacks RNAIII, the effector molecule of *agr*-mediated phenotypes with respect to $H_2O_2$. (**B**) Trans-activation demonstrated by hemolysin activity on sheep blood agar plates. Bottom of figure shows zone of clearing (hemolysin activity) after mixing $10^8$ Δ*agrBD* CFU with an equal number of Δ*rnaIII*. Zone of clearance is a consequence of AgrC receptor activation in trans by AIP produced by the Δ*rnaIII* mutant. (**C**) Absence of trans-activation with short-term culture. The wild-type strain RN6734 (WT, BS435), Δ*rnaIII* (GAW183), Δ*agrBD* (GAW130), and Δ*rnaIII* and Δ*agrBD* mutants were mixed 1:1 immediately before growth from overnight culture. Overnight cultures were diluted ($OD_{600}$~0.05) into fresh Tryptic Soy Broth (TSB) medium, mixed, and grown to early log phase ($OD_{600}$~0.15) when they were treated with 20 mM $H_2O_2$ for 60 min and assayed for percent survival by plating. (**D**) Kinetics of killing by $H_2O_2$. Survival assays employing Δ*rnaIII* and Δ*agrBD* mixtures, performed as in panel C, but

*Figure 3 continued*

grown from early exponential (1 hr, OD$_{600}$~0.15) through late log (5 hr, OD$_{600}$~4) phase in TSB. Cultures were treated with H$_2$O$_2$ (20 mM for 1 hr) at the indicated time points. (**E**) Proportion of mixed population for panel D represented by each mutant after incubation. The Δ*agrBD* mutant contained an erythromycin-resistance marker to distinguish the strains following plating of serial dilutions on TS agar with or without erythromycin (5 µg/). Data represent the mean ± SD. from biological replicates (n=3). (**F**) Trans-activation during long-term culture. The wild-type strain RN6734 (WT, BS435), Δ*rnaIII* (strain GAW183), Δ*agrBD* (strain GAW130), and Δ*rnaIII* and Δ*agrBD* mutants mixed 1:1 prior to overnight culture. Survival assays employing Δ*rnaIII* and Δ*agrBD* mixtures, performed as in panel C. (**G**) Kinetics of killing by H$_2$O$_2$. Survival assays employing Δ*rnaIII* and Δ*agrBD* mixtures, performed as in panel D. Cultures were treated with H$_2$O$_2$ (20 mM for 1 hr) at the indicated time points. (**H**) Proportion of mixed population for panel G represented by each mutant after incubation, performed as in panel E. Data represent the mean ± SD. from biological replicates (n=3).

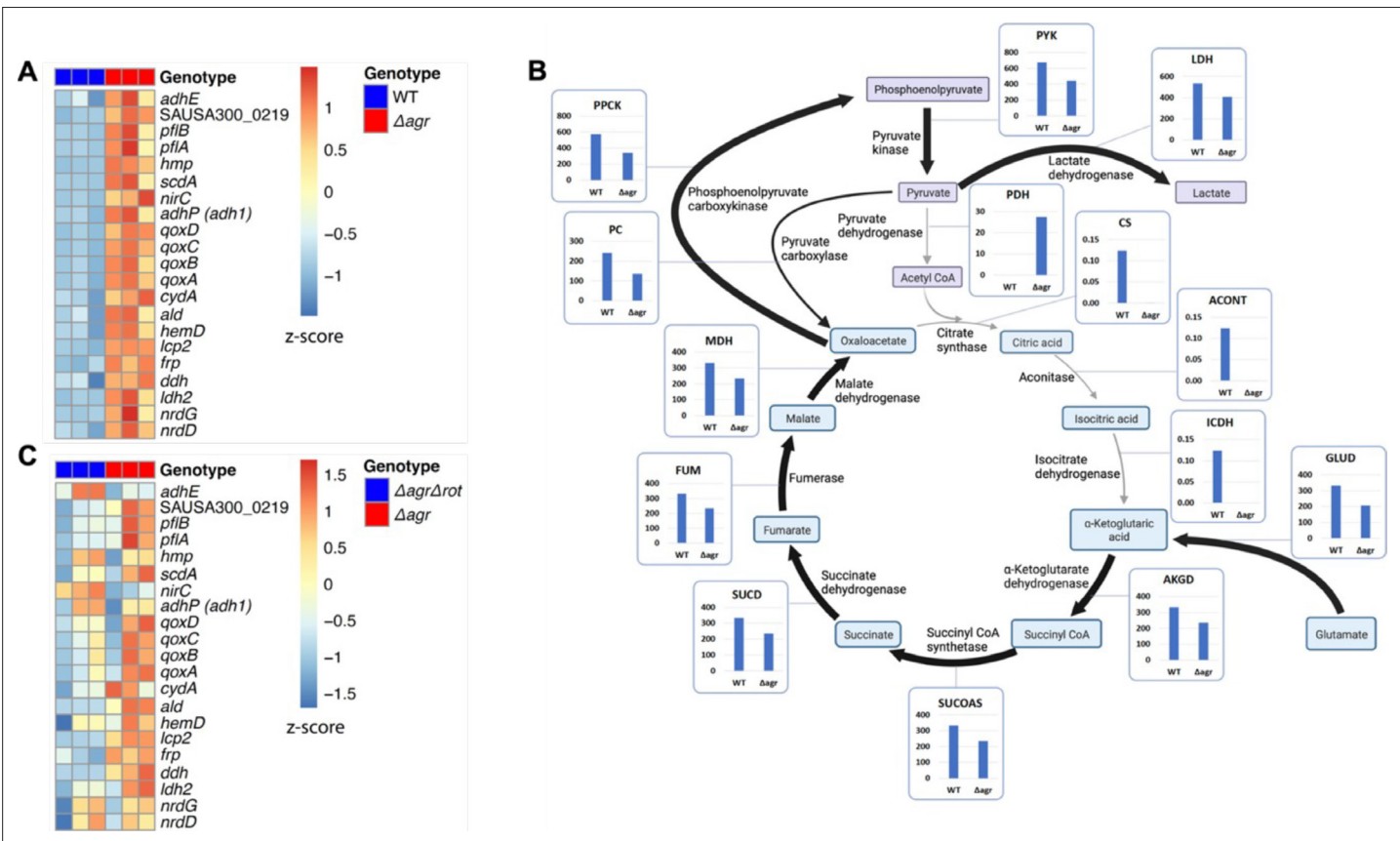

**Figure 4.** Association of *agr* deficiency with increased expression of respiration and fermentation genes during aerobic growth. (**A**) Relative expression of respiration and fermentation genes. RNA-seq comparison of *S. aureus* LAC wild-type (WT, BS819) and Δ*agr* mutant (BS1348) grown to late exponential phase (OD$_{600}$~4.0). Shown are significantly up-regulated genes in the Δ*agr* mutant (normalized expression values are at least twofold higher than in the wild-type). Heatmap colors indicate expression z-scores. RNA-seq data are from three independent cultures. See ***Supplementary file 1*** for supporting information. (**B**) Schematic representation of *agr*-induced changes in metabolic flux, inferred from transcriptomic data (***Supplementary file 1***) by SPOT (Simplified Pearson correlation with Transcriptomic data). Metabolic intermediates and enzymes involved in catalyzing reactions are shown. The magnitude of the flux (units per 100 units of glucose uptake flux) is denoted by arrowhead thickness. Boxed charts indicate relative flux activity levels in wild-type versus Δ*agr* strains. Enzyme names are linked to abbreviations in boxed charts (e.g. lactate dehydrogenase, LDH). See ***Supplementary file 2*** for supporting information. (**C**) RNA-seq comparison of an Δ*agr* Δ*rot* double mutant (BS1302) with its parental Δ*agr* strain (BS1348). Heatmap colors indicate expression z-scores. Sample preparation and figure labeling as for **A**. See ***Supplementary file 3*** for supporting information.

The online version of this article includes the following figure supplement(s) for figure 4:

**Figure supplement 1.** Induction of expression of selected fermentive/anaerobic genes stimulated by deletion of *agr*.

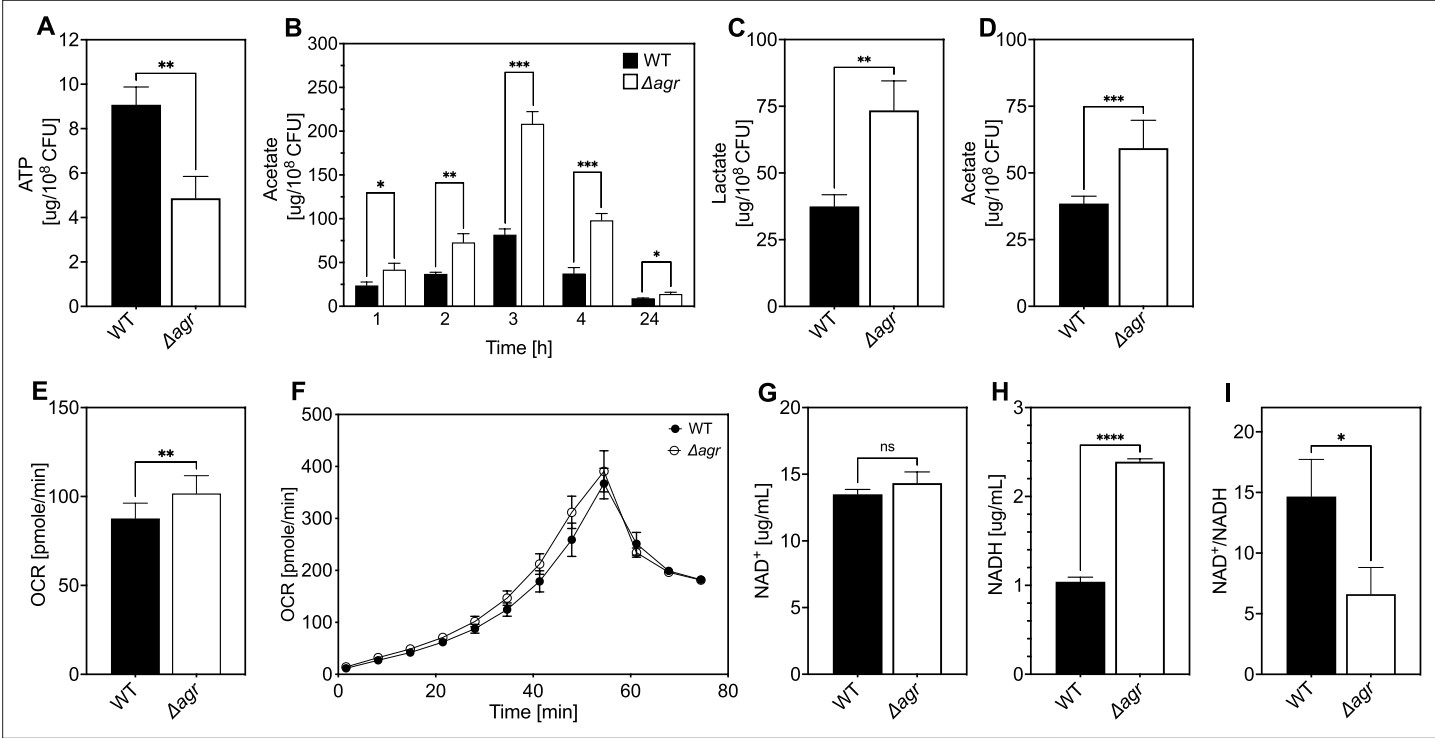

**Figure 5.** Association of *agr* deficiency with a metabolic flux shift toward fermentive metabolism during aerobic growth. (**A**) Intracellular ATP levels. Comparison of *S. aureus* LAC wild-type (WT, BS819) and Δ*agr* mutant (BS1348) strains for ATP expressed as µg/10⁸ cells after growth of cultures in Tryptic Soy Broth (TSB) medium to late-exponential phase (OD$_{600}$~4.0). (**B**) Extracellular acetate levels. Samples were taken after 1, 2, 3, 4, and 24 hr of growth in TSB medium; strains were wild-type (WT, BS819) and Δ*agr* mutant (BS1348). (**C–D**) Extracellular lactate and acetate levels during low oxygen culture. *S. aureus* LAC wild-type (WT, BS819) and Δ*agr* mutant (BS1348) were grown in TSB medium with suboptimal aeration to late-exponential phase (4 hr, OD$_{600}$~4.0). (**E–F**) Oxygen consumption. Strains LAC wild-type (WT, BS819) and Δ*agr* mutant (BS1348) were compared using Seahorse XFp analyzer (**F**), and the rate of oxygen consumption (**E**) was determined from the linear portion of the consumption curve. Representative experiments from at least three independent assays are shown. (**G–H**) NAD⁺ and NADH levels. Colorimetric assay of NAD⁺ (**G**) and NADH levels (**H**) for *S. aureus* wild-type (WT, BS819) and Δ*agr* mutant (BS1348) after growth of cultures to late-exponential phase (OD$_{600}$~4.0). (**I**) NAD⁺/NADH ratio. For all panels, data points are the mean value ± SD (n=3). *$p<0.05$; ****$p<0.0001$, by Student's two-tailed *t*-test. Seahorse statistical significances are compared to TSB medium.

The online version of this article includes the following figure supplement(s) for figure 5:

**Figure supplement 1.** Association of *agr* deficiency with glucose consumption and intracellular levels of pyruvate, acetyl-CoA, and TCA-cycle metabolites.

Δ*agr* mutants exhibit TCA cycle proficiency (*Somerville et al., 2002a*) and, despite some expense of efficiency, an increased catabolism of acetate.

Differential transcription of selected genes was confirmed by RT-qPCR measurements (*Figure 4— figure supplement 1*) We also confirmed that respiration levels were lower (15%) in wild-type compared to Δ*agr* (*Figure 5E, F*). Although the stimulatory effect of the *agr* deletion on production of the fermentation product lactate was not observed in optimally aerated broth cultures after growth to late exponential growth phase, it was confirmed for organisms grown in broth under more metabolically demanding suboptimal aeration conditions (limitations in the rate of respiration when oxygen is limiting are expected to increase overall levels of fermentation) (*Figure 5C*). Overall, these results are consistent with transcription-level up-regulation of respiratory and fermentative pathways in *agr*-deficient strains.

Since respiration and fermentation generally increase NAD⁺/NADH ratios and since these activities are increased in Δ*agr* strains (*Figure 5C and E–F*), we expected a higher NAD⁺/NADH ratio relative to wild-type cells. However, we observed a decrease in the NAD⁺/NADH ratio due to an increase in NADH accompanied by relative stability in NAD⁺ compared to wild-type. Collectively, these observations suggest that a surge in NADH accumulation and reductive stress in the Δ*agr* strain induces a burst in respiration, but levels of NADH are saturating, thereby driving fermentation under microaerobic conditions.

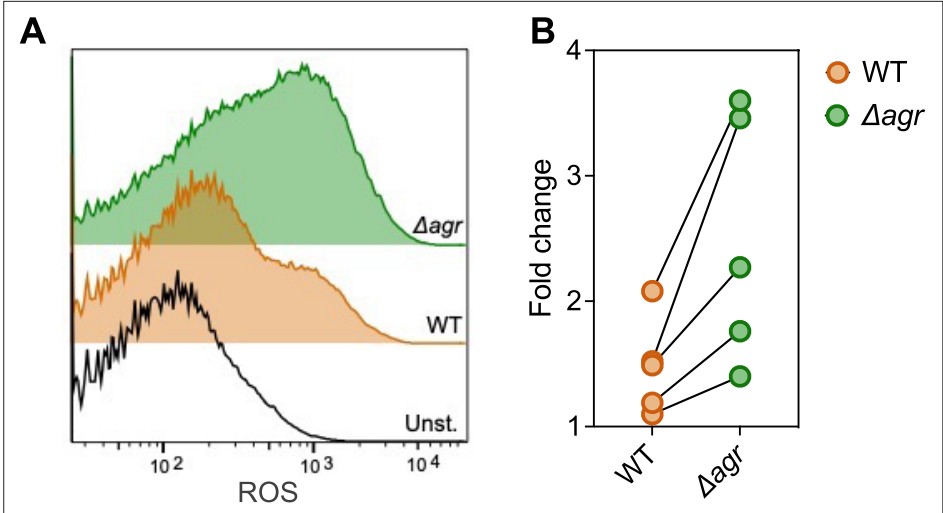

**Figure 6.** Increase in reactive oxygen species (ROS) levels associated with Δ*agr* deficiency. Flow cytometry measurements. *S. aureus* LAC wild-type (WT, BS819) and Δ*agr* mutant (BS1348) were grown overnight, diluted, cultured in Tryptic Soy Broth (TSB) medium for 1 hr, and treated with carboxy-H2DCFDA (10 μM) for 5 min. Relative cell number is on the vertical axis. Unst. indicates samples containing LAC wild-type cells not treated with carboxy-H2DCFDA. (**B**) Five replicate experiments gave similar results ('fold change' indicates the mean wild-type or Δ*agr* ROS level divided by the mean autofluorescence background signal; lines connect results in replicate experiments).

To help determine the metabolic fate of glucose, we measured glucose consumption and intracellular levels of pyruvate and TCA-cycle metabolites fumarate and citrate in the wild-type and Δ*agr* mutant strains. At 4 hr of growth to late-exponential phase, intracellular pyruvate, and acetyl-CoA levels were increased in the Δ*agr* mutant compared to wild-type strain, but levels of fumarate and citrate were similar (*Figure 5—figure supplement 1D–E*). Glucose was depleted after 4 hr of growth, but glucose consumption after 3 hr of growth (exponential phase) was increased in the Δ*agr* mutant compared to the wild-type strain (*Figure 5—figure supplement 1A*). These observations, together with the decrease in the $NAD^+$/NADH ratio and increase in acetate and lactate production described above, are consistent with a model in which respiration in Δ*agr* mutants is inadequate for (1) energy production, resulting in an increase in acetogenesis, and (2) maintenance of redox balance, resulting in an increase in fermentative metabolism, lactate production, and conversion of NADH to $NAD^+$. Increased levels of acetate compared to lactate under optimal aeration conditions suggests that demand for ATP is in excess of demand for $NAD^+$.

Elevated respiratory activity of Δ*agr* is expected to increase endogenous ROS (*Lobritz et al., 2015*). To test this idea, we assessed ROS accumulation in bulk culture by flow cytometry of Δ*agr* and wild-type stains using carboxy-H2DCFDA, a dye that becomes fluorescent in the presence of several forms of ROS. As shown in *Figure 6*, ROS levels increased with *agr* deficiency, indicating correlation between *agr* activity, lower ROS levels, and increased bacterial survival in response to exogenous $H_2O_2$. These data help explain the elevated lethality of peroxide in the absence of *agr*. Since lower ROS accumulation in wild-type cells correlates with decreased respiration and protection from killing by $H_2O_2$, the data also support the idea that suppression of endogenous ROS is key to *agr*-mediated protection from exogenous $H_2O_2$-mediated lethality.

### Transcriptional changes due to Δ*agr* mutation are long-lived and result in down-regulation of $H_2O_2$-stimulated genes relative to those in an agr wild-type

We reasoned that the transcriptional changes due to the Δ*agr* mutation likely persist, as does this strain's susceptibility to killing by $H_2O_2$, after growth from overnight culture. With this in mind, and to determine whether *agr*-mediated changes act through *rot*, we performed RNA-seq experiments after 1 hr growth from overnight cultures of a Δ*agr* Δ*rot* double mutant that phenocopies wild-type with

respect to H₂O₂-mediated death and with respect to its parental Δ*agr* strain (***Supplementary file 3***). Fold-changes and number of genes differentially expressed were lower in the Δ*agr* mutant relative to the wild-type culture, potentially because a significant portion of the population, even after an hour of growth (early exponential phase), still consisted of cells experiencing stationary phase at the time of sampling. Nevertheless, we did observe a shift in the expression of fermentation-associated genes (*ilvA, pflAB, aldh1, ddh, lctp2*) in the Δ*agr* strain (***Figure 4C*** and ***Supplementary file 3***). Thus, up-regulation of metabolic genes in the Δ*agr* mutant extends beyond post-exponential growth to the exit from stationary phase and into subsequent cell proliferation, as does the long-lived protection from H₂O₂-mediated killing seen with the wild-type strain.

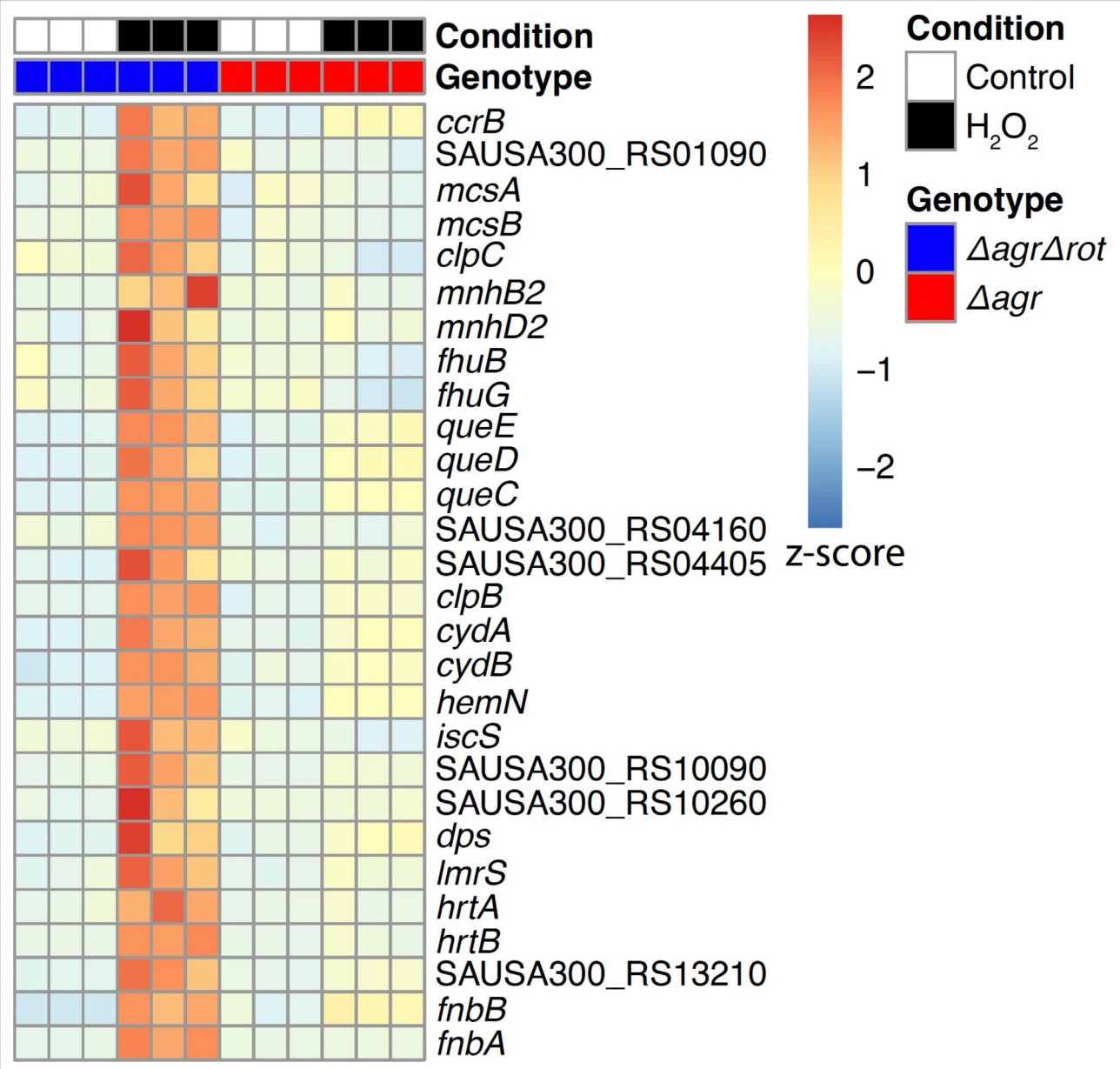

**Figure 7.** Rot-mediated up-regulation of H₂O₂-stimulated genes relative to those in an *agr* mutant. Genes shown are those up-regulated in a Δ*agr* Δ*rot* double mutant (BS1302) relative to that observed with the Δ*agr* strain (BS1348). H₂O₂ treatment was for 30 min. Peroxide concentrations for Δ*agr* (2.5 mM H₂O₂) and Δ*agr* Δ*rot* (10 mM H₂O₂) were determined to achieve ~50% cell survival [see Methods and ***Figure 7—figure supplement 1***]. RNA-seq data are from three independent cultures. Heatmap colors indicate expression z-scores. See ***Supplementary file 3*** for supporting information.

The online version of this article includes the following figure supplement(s) for figure 7:

**Figure supplement 1.** Normalization of the lethal concentration of H₂O₂ with wild-type and Δ*agr* strains.

To examine the induction of genes by lethal levels of $H_2O_2$, our gene expression analysis included a comparison between untreated and $H_2O_2$-treated cells after growth from overnight culture (**Supplementary file 3**). The Δagr Δrot double mutant that phenocopies wild-type had elevated expression of many genes involved in lowering oxidative stress compared to the Δagr mutant. Those genes are involved in the regulation of misfolded proteins (mcsA, mcsB, clpC, clpB), Fe-S cluster repair (iscS), DNA protection and repair (dps), and genes regulated by the protein-damage repair gene bshA (fhuB/G, queC-E) (**Posada et al., 2014**; **Figure 7**, **Figure 7—figure supplement 1**, and **Supplementary file 3**). Elevated expression of protective genes suggests that the double mutant survives damage from $H_2O_2$ better because protective genes are rendered inducible (loss of Rot-mediated repression). Overall, the data show that agr wild-type cells assume a long-lived stage after activation at high cell density in which they are primed to express genes (e.g. clpB/C, dps) that protect against high levels of exogenous oxidative stress.

## Endogenous ROS is involved in agr-mediated protection from lethal, exogenous $H_2O_2$ stress

We next monitored the effect of reducing respiration and ATP levels by adding subinhibitory doses of the redox cycling agent menadione (**Rowe et al., 2020**) to cultures of Δagr and wild-type cells prior to lethal levels of $H_2O_2$. Addition of menadione for 30 min, which induces a burst of ROS that inactivates the TCA cycle and thereby respiration (**Rowe et al., 2020**), protected the Δagr mutant but had little effect on the wild-type strain (**Figure 8A**). Menadione's effect on respiration and ATP can be reversed by N-acetyl cysteine (**Rowe et al., 2020**). Addition of N-acetyl cysteine in the presence of menadione restored $H_2O_2$ susceptibility to the agr mutant (**Figure 8A**). Thus, blocking endogenous ROS production/accumulation reverses the lethal effect of an agr deficiency with respect to a subsequent exogenous challenge with $H_2O_2$.

**Rowe et al., 2020** showed that menadione exerts its effects on endogenous ROS by inactivating the TCA cycle in *S. aureus*. To determine whether this mechanism can induce protection in the Δagr mutant, we inactivated the TCA cycle gene acnA in agr wild-type and Δagr strains (**Figure 8—figure supplement 2**). We found that ΔacnA mutation completely protected the Δagr mutant from peroxide killing after growth to late exponential growth phase but had little effect on the wild-type agr strain. This finding supports the idea that TCA cycle activity contributes to an imbalance in endogenous ROS homeostasis in the Δagr mutant, and that this shift is a critical factor for Δagr hyperlethality. When we evaluated long-lived protection by comparing survival rates of Δagr ΔacnA mutant and Δagr cells following dilution of overnight cultures and regrowth prior to challenge with $H_2O_2$, ΔacnA remained protective, but less so (**Figure 8—figure supplement 2**). These partial effects of an ΔacnA deficiency suggest that Δagr stimulates long-lived lethality for peroxide through both TCA-dependent and TCA-independent pathways.

*S. aureus* has multiple enzymes that control the endogenous production and detoxification of ROS. SodA and SodM dismutate superoxide ($O_2^{\bullet-}$) to $H_2O_2$, and catalase and AhpC then convert $H_2O_2$ to water, limiting the formation of toxic hydroxyl radical ($OH^\bullet$). Accordingly, we asked whether mutations in these pathways affect agr-dependent phenotypes with respect to lethal $H_2O_2$ exposure. A deficiency in the sodA superoxide dismutase (**Clements et al., 1999**) resulted in lower survival of the wild-type strain, similar to that observed with the Δagr mutant (**Figure 8B–C**). *The effect was reversed by* complementation with sodA on a low-copy-number plasmid (**Figure 8D**). The ΔsodA mutation had no effect on killing with the Δagr strain. Moreover, sodA expression (**Supplementary file 1**) and activity levels (**Figure 8E**) were similar for wild-type and the Δagr mutant. Together, these observations suggest that the contribution of sodA toward protective priming by wild-type involves dismutation of low levels of endogenous superoxide generated by respiration. In contrast, endogenous levels of ROS are saturating for sodA in Δagr cells. Inactivation of sodM, which is thought to be primarily induced by exogenous oxidative stress (**Gaupp et al., 2012**), had no noticeable effect on the $H_2O_2$ susceptibility of the wild-type or the Δagr mutant. We conclude that scavenging enzymes, such as SodA, are better able to control the threat posed by endogenous ROS in wild-type than in Δagr cells. They render the former better able to survive a subsequent lethal dose of $H_2O_2$, a compound that freely enters cells (**Imlay, 2008**) and would add to endogenous ROS levels.

Other oxidative-stress-response mutations in genes encoding catalase, thiol-dependent peroxidases, and bacillithiol showed little effect on the relative lethality of $H_2O_2$ between wild-type and Δagr

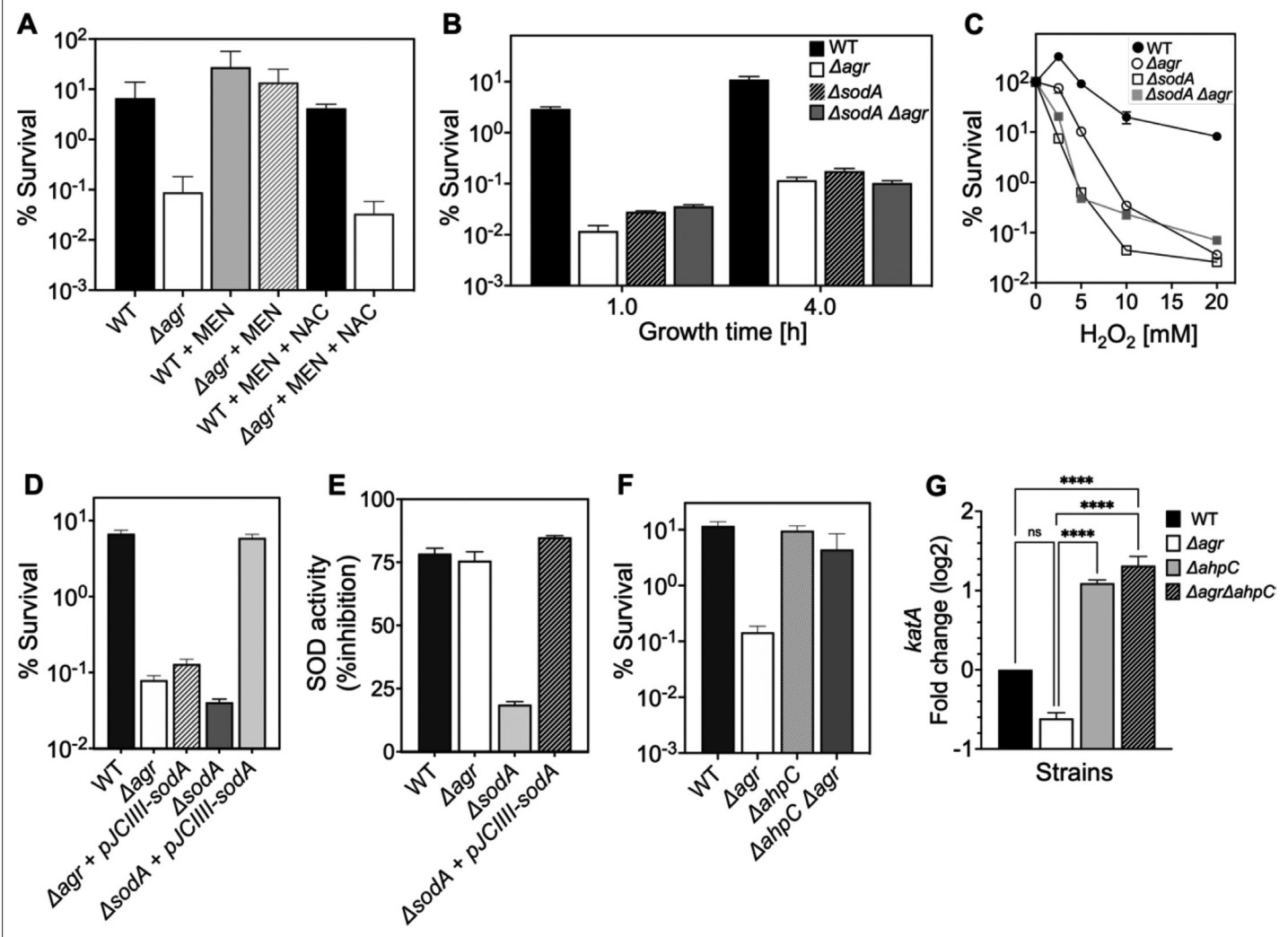

**Figure 8.** Involvement of endogenous reactive oxygen species (ROS) in *agr*-mediated protection from lethal $H_2O_2$ stress. (**A**) Protective effect of menadione on survival. *S. aureus* LAC wild type (BS819) and Δ*agr* mutant (BS1348) cultures were grown to late exponential phase (4 hr after dilution of overnight cultures), exposed to 80 μM menadione (MD) with or without 4 mM N-acetyl cysteine (NAC) for 30 min prior to treatment with $H_2O_2$ (20 mM for 1 hr) and measurement of survival. (**B**) Effect of *sodA* deletion on survival. Cultures of wild-type (BS819), Δ*agr* (BS1348), a *sodA::tetM* (BS1422), *and sodA::tetM-agr* double mutant (BS1423) were grown to early (1 hr after dilution, $OD_{600}$~0.15) or late log (4 hr after dilution, $OD_{600}$~4.0) prior to treatment with 20 mM $H_2O_2$ for 60 min. (**C**) Effect of $H_2O_2$ concentration on survival. Late log (4 hr, $OD_{600}$~4.0) cultures of the wild-type and Δ*agr* mutant strains were treated with indicated concentrations of $H_2O_2$ for 60 min. (**D**) Complementation of *sodA* deletion mutation. A plasmid-borne wild-type *sodA* gene was expressed under control of the *sarA* constitutive promoter (pJC1111-*sodA*) in late log-phase (4 hr, $OD_{600}$~4.0) cells treated with 20 mM $H_2O_2$ for 60 min. (**E**) SodA activity. Wild-type or the indicated mutants were grown to late-exponential phase ($OD_{600}$~4.0); Sod activity was measured as in Methods. (**F**) Effect of *ahpC* deletion on survival. Late log-phase cultures of wild-type (BS819), Δ*agr* (BS1348), *ahpC::bursa* (BS1486), and Δ*ahpC::bursa-agr* double-mutant (BS1487) cells were treated with 20 mM $H_2O_2$ for 60 min. (**G**) Effect of *ahpC* deletion on expression of *katA* in the indicated mutants. Total cellular RNA was extracted from late exponential-phase cultures ($OD_{600}$~4.0), followed by reverse transcription and PCR amplification of the indicated genes, using *rpoB* as an internal standard. mRNA levels were normalized to those of each gene to wild-type control. Data represent the mean ± SD. from (n=3) biological replicates. One-way ANOVA was used to determine statistical differences between samples (****p<0.0001).

The online version of this article includes the following figure supplement(s) for figure 8:

**Figure supplement 1.** Deficiency in reactive oxygen species (ROS) detoxification genes *katA, bsaA1/gpxA1, bsaA2/gpxA2*, and bacilliothiol (BSH) have no effect on *agr*-mediated protection from $H_2O_2$-mediated cell death.

**Figure supplement 2.** Deficiency in TCA cycle gene *acnA* reverses the effect of an *agr* deficiency with respect to subsequent challenges with $H_2O_2$.

**Figure supplement 3.** Effects of transposon insertion in *ahpC* unexplained by polarity of transposon insertion.

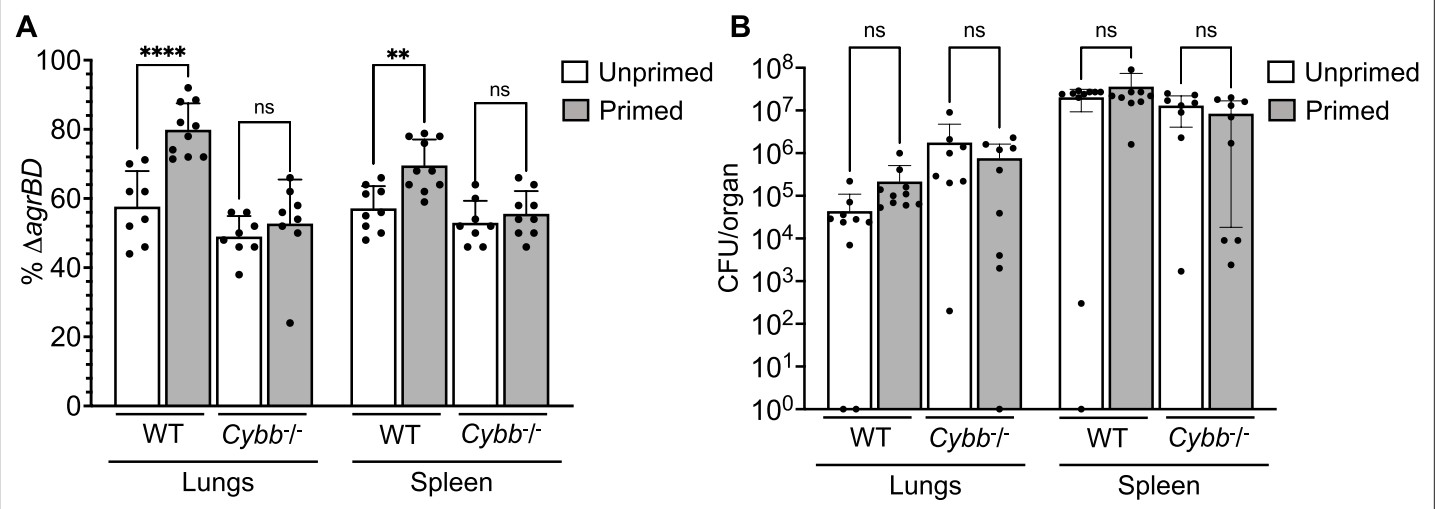

**Figure 9.** Survival advantage of *agr* priming of *S.aureus* absent in phagocyte NADPH-deficient murine infection. (**A**) percentage of Δ*agrBD* (AIP-responsive in-frame deletion mutant carrying an intact RNAIII) cells and (**B**) bacterial burden in lung or spleen after 2 hr of intraperitoneal infection of wild-type (WT) C57BL/6 mice or phagocyte NADPH oxidase-deficient (*Cybb⁻/⁻*) mice (see *Figure 9—figure supplement 1* for data with other organs). Δ*agrBD* and Δ*rnaIII* mutant cultures were grown separately and mixed at a 1:1 ratio either before (primed) or after (unprimed) overnight growth, as for *Figure 3*. Both primed and unprimed mixtures were diluted after overnight growth, grown to early log phase ($OD_{600} \sim 0.15$), and used as inocula ($1 \times 10^8$ CFU) for intraperitoneal infection (n=2 groups of 10 mice each). After 2 hr, lungs and spleen were harvested and homogenized; aliquots were diluted and plated to enumerate viable bacteria. Output ratios and total and mutant colony forming units (CFU) from tissue homogenates were determined as for *Figure 4E and H*. A Mann-Whitney test (panel 9 A) or Student's two-tailed *t*-test (panel 9B) were used to determine the statistical significance of the difference between primed and unprimed cultures. Error bars indicate standard deviation ($**p<0.01$; $****p<0.0001$).

The online version of this article includes the following figure supplement(s) for figure 9:

**Figure supplement 1.** Long-lived protection by *agr* increases peritoneal fitness and dissemination to liver, kidney, and heart in both C57BL/6 mice and C57BL/6 *Cybb⁻/⁻* (*gp91phox/nox2*) mice.

mutant strains (*Figure 8—figure supplement 1*). Thus, protection against $H_2O_2$ lethality by these genes is not *agr*-specific. Paradoxically, a deficiency of *ahpC* (*ahpC::bursa*), which encodes a peroxidase (*Cosgrove et al., 2007*), almost completely reversed the elevated killing associated with the Δ*agr* mutation (*Figure 8F*). An *ahpC* deficiency had no effect on the response of the otherwise wild-type strain. A deficiency in other downstream genes in the *ahpC* operon (*ahpF*, SAUSA300_0377–0378) showed no effect, indicating that the protective behavior of mutant *ahpC* was not caused by polar effects (*Figure 8—figure supplement 3*).

Results with *ahpC* deficiencies were initially surprising, because reduced ROS detoxification should increase rather than decrease killing. Compensatory expression of other protective genes, such as *katA* in the Δ*ahpC* Δ*agr* double mutant (*Cosgrove et al., 2007*), might enable cells to better survive damage from subsequent stress-stimulated ROS increases. Indeed, *katA* expression increased >10 fold in the Δ*ahpC* and Δ*ahpC* Δ*agr* double mutants (*Figure 8G*). Thus, Δ*katA* overcomes Δ*ahpC*-mediated protection, consistent with the idea expressed previously that *katA* is more protective than *ahpC* against high levels of exogenous oxidative stress (*Cosgrove et al., 2007*; *Seaver and Imlay, 2001*). We conclude that the protective action of an *ahpC*-deficient mutant is due to a pre-induced, compensatory increase in the expression of another protective catalase.

### Importance of the long-lived 'memory' of agr-mediated protection in a murine intraperitoneal infection model

To determine whether long-lived *agr*-mediated protection is important for *S. aureus* pathogenesis, we used the mixed infection strategy (outlined in *Figure 3*) in which a Δ*agrBD* mutant is 'primed' in response to AIP produced by a Δ*rnaIII* mutant after overnight co-culture containing an equal ratio of the two mutant strains (*Figure 9A* and *Figure 9—figure supplement 1*). Then mice were infected via intraperitoneal inoculation; 2 hr later, we lavaged the peritoneal cavity and harvested organs for determination of colony forming units (CFU). By 2 hr after bacterial administration, the number of *S.*

*aureus* cells injected as inoculum had declined by 1000-fold (*Figure 9B* and *Figure 9—figure supplement 1*). Mutant proportions, identified by differential plating, demonstrated that Δ*agrBD* cells were enriched by ~30% in both peritoneum and organs compared to the Δ*rnaIII* mutant. The fraction of Δ*agrBD* (*rnaIII*⁺) mutants in sites of bacterial dissemination (heart, kidney, liver, lungs, and spleen) was similar to their elevated fraction in the peritoneum, thereby suggesting that *agr* enhances intraperitoneal infection and access to, rather than entry into extraperitoneal organs. In a control infection in which Δ*agrBD* was 'unprimed' by mixing Δ*agrBD* and Δ*rnaIII* mutants immediately before growth from stationary phase, the proportion of Δ*agrBD* bacterial burden was lower at all tissue sites (*Figure 9A* and *Figure 9—figure supplement 1*). This drop represented a decline in long-lived *agr* induction of virulence.

To study *agr*-ROS effects during infection, we repeated in vivo studies using *Cybb*⁻/⁻ mice deficient in enzymes associated with host phagocyte production of ROS (the gp91 [phox] component of the phagocyte NADPH oxidase)(*Pollock et al., 1995*). We found that *agr*-mediated priming (mixing Δ*agrBD* and Δ*rnaIII* before overnight co-culture) failed to increase hematogenous dissemination to lung and spleen tissues following infection of *Cybb*⁻/⁻ mice (*Figure 9*). Thus, when the host makes little ROS, long-lived **agr**-mediated protection has little effect in these tissues. The data also indicate that *agr*-mediated protection against **ROS** enhances fitness in lung and spleen, but it is dispensable for full virulence in other organs. Collectively, the murine experiments indicate that the long-lived 'memory' of *agr* induction enhances overall pathogenicity of *S. aureus* during sepsis. They also support data previously published indicating that clearance of disseminating bloodstream pathogens (*Yipp et al., 2017*) and protection from ROS buildup (*Beavers et al., 2021*) are tuned to particular sites in the host organism.

## Discussion

We report that *agr*, a quorum-sensing regulator of virulence in *S. aureus*, provides surprisingly long-lived protection from the lethal action of exogenous $H_2O_2$. The protection, which is uncoupled from *agr* activation kinetics, arises in part from limiting the accumulation of endogenous ROS. This apparent tolerance to lethal stress derives from an RNAIII-*rot* regulatory connection that couples virulence-factor production to metabolism and thereby to levels of ROS. Collectively, the results suggest that *agr* anticipates and protects the bacterium from increases in ROS expected from the host during *S. aureus* infection.

Details of *agr*-mediated protection are sketched in *Figure 10*. At low levels of ROS, *agr* is activated by a redox sensor in AgrA, RNAIII is expressed and represses the Rot repressor, thereby rendering protective genes (e.g. *clpB/C, dps*) inducible via an unknown mechanism (induction, candidate protective gene(s), and their connection to endogenous ROS levels are being pursued, independent of the current report). Superoxide dismutase and scavenging catalases/peroxidases detoxify superoxide and peroxide, respectively (scavenging deficiencies reduce the protective effect of wild-type). Deletion of *agr* eliminates expression of RNAIII and repression of Rot, resulting in a metabolic instability associated with a 100-fold increase in $H_2O_2$-mediated death.

The *agr* system directly reduces $H_2O_2$-mediated killing by reducing levels of endogenous ROS, much like intrinsic tolerance to lethal antimicrobial stress (*Zeng et al., 2022*). However, the protective system we describe is distinct in that it primes cells for induction of genes (e.g. *clpB/C, dps*) that mitigate damage upon subsequent exposure to high levels of ROS. Still unidentified protective genes exist; thus, *agr*-mediated protection may be further shaped by both known (*ahpC*) and unidentified pathways and factors that modify the redox state. Another distinctive feature of *agr*-mediated protection is its manifestation even in early log-phase cultures, long after the maximal transcription of *agr* at high cell density, i.e., quorum. In a sense, *S. aureus* has a 'memory' of the *agr*-activated state.

Transcriptional profiling during growth from diluted, overnight cultures revealed that the Δ*agr* mutation elevated the expression of several respiration and fermentation genes. Acceleration of cellular respiration is likely a source of ROS, as it appears to be for bactericidal antibiotics (*Kohanski et al., 2007*). Our work supports this idea by showing that increased respiration caused by deletion of *agr* is associated with increased ROS-mediated lethality. How *agr* deficiency is connected to the corruption of downstream processes that result in metabolic inefficiency and increased endogenous ROS levels is unknown. Given that Δ*agr* mutants are unable to downregulate surface proteins during stationary phase (*Morfeldt et al., 1995*; *Novick et al., 1993*), it is possible that deletion of *agr* perturbs the

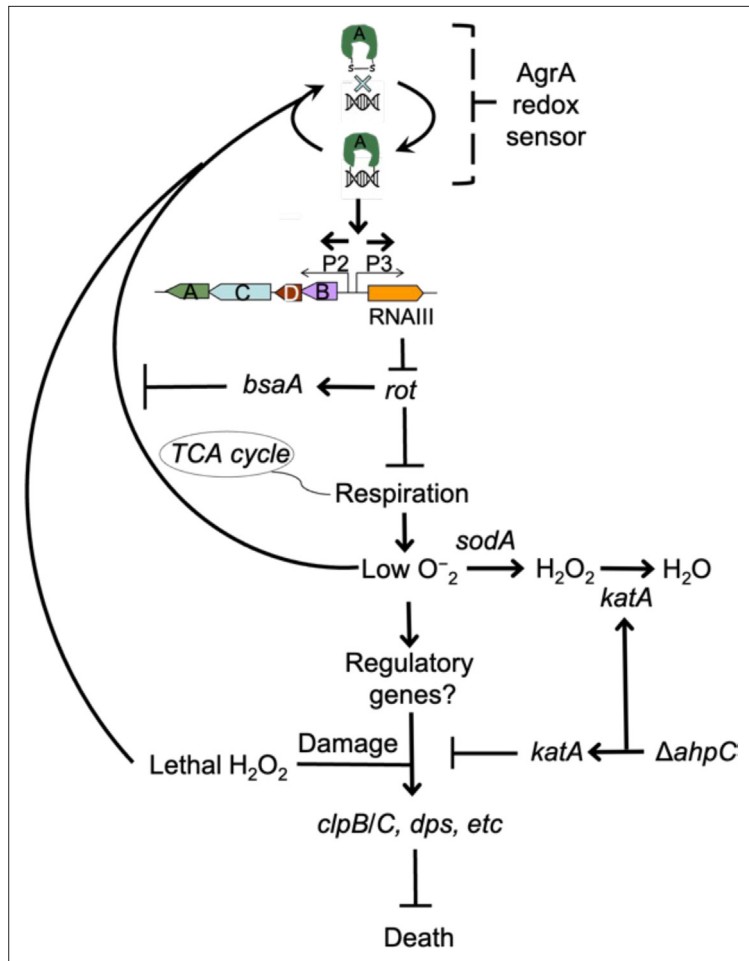

**Figure 10.** Schematic representation of *agr*-mediated protection from reactive oxygen species (ROS). At low levels of oxidative stress, the redox sensor in AgrA binds to DNA at promoters P2 and P3, activating expression of the two operons. Expression of RNAIII blocks translation of Rot, which decreases respiration and production of superoxide. ROS quenchers (*sodA* and *katA/ahpC*) suppress formation of most ROS that would otherwise signal the redox sensor in AgrA to halt stimulation of RNAIII expression and the production of further superoxide via respiration. This feedback system regulates respiration thereby limiting the accumulation of ROS in wild-type cells. Wild-type cells are primed for induction of protective genes (e.g. *clpB/C, dps*) by loss of the *rot* repressor system via an unknown mechanism when cells experience damage from high levels of oxidative stress (experimentally introduced as lethal exogenous $H_2O_2$); $\Delta agr$ cells that experience high levels of endogenous $H_2O_2$ fail to induce protective genes. Exogenous $H_2O_2$ or high levels of endogenous ROS, for example from extreme stress due to ciprofloxacin (***Kumar et al., 2017***), lower RNAIII expression and allow Rot to stimulate *bsaA* expression, which produces a protective antioxidant. The protective action of an *ahpC* deficiency acts through compensatory expression of *katA*, which results in more effective scavenging of $H_2O_2$ produced from increased respiration in $\Delta agr$ strains and/or exogenous lethal $H_2O_2$.

---

cytoplasmic membrane or the machinery that sorts proteins across the cell wall. In support of this notion, jamming SecY translocation machinery of *E. coli* results in downstream events shared with antibiotic lethality, including accelerated respiration and accumulation of ROS (***Takahashi et al., 2017***). In this scenario, the formation of a futile macromolecular cycle may accelerate cellular respiration to meet the metabolic demand of unresolvable problems caused by elevated surface sorting.

As noted above, *agr* is inactivated by oxidation, which elevates levels of the antioxidant BsaA during exposure to $H_2O_2$ (***Sun et al., 2012***). That would make our finding that $H_2O_2$-mediated killing is increased in the $\Delta agr$ mutant paradoxical. This apparent inconsistency can be explained by a focus

of prior work on growth-related phenotypes (*Sun et al., 2012*) rather than on lethality (the underlying mechanisms are distinct *Drlica and Zhao, 2021*). Additionally, we note that *bsaA* was not upregulated in either our RNA-seq experiments (*Supplementary file 1*) or in previous transcriptional profiling data (*George et al., 2019*). Thus, an alternative, but not mutually exclusive, hypothesis is that the growth-related effect of *bsaA* on *agr*-mediated responses to stress is strain-dependent. Another complexity involves test conditions, as indicated by consideration of previous work in which wild-type cells exhibited greater oxidative stress than the *agr*-deficient mutant due to *agrA*-mediated production of ROS-inducing phenol-soluble modulins (*George et al., 2019*). The present experiments were performed in highly diluted cultures in which levels of these modulins are likely low (*Queck et al., 2008*; *Wang et al., 2007*). The complex relationship between *agr*, ROS-mediated lethality, and physiological state illustrates the importance of understanding *agr* biology before applying therapies that inactivate *agr* (*Khan et al., 2015*).

We also note that although the absence of *agr* increases killing by high levels of $H_2O_2$, it has the opposite effect on lethal concentrations of ciprofloxacin (*Kumar et al., 2017*). In the latter case, the absence of *agr* upregulates the expression of *bsaA* in the strain examined; *bsaA* counters endogenous ROS induced by ciprofloxacin (*Kumar et al., 2017*). The present work shows that excess endogenous

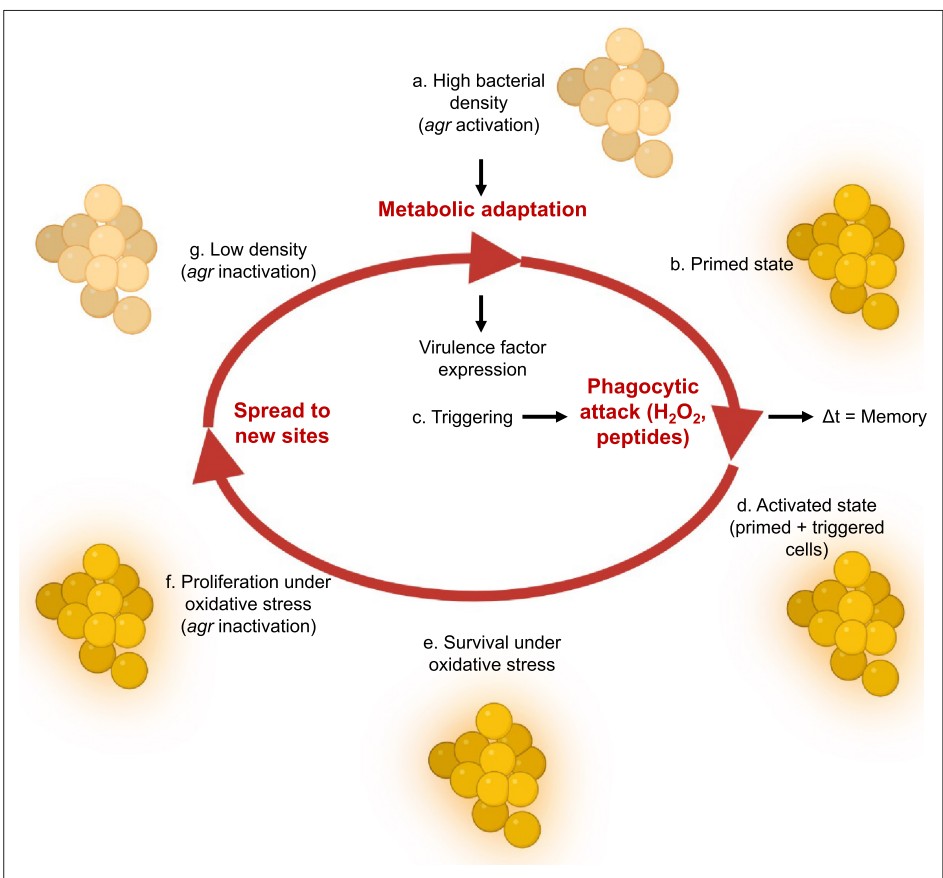

**Figure 11.** Relationship of *agr* priming and virulence. The ecology of abscess formation and subsequent bacterial dissemination can be described as a cycle. (**a**) During abscess formation, a hallmark of *S. aureus* disease, *agr* is activated by high bacterial cell density (quorum sensing) (*Wright et al., 2005*). (**b**) The bacterium assumes a primed stage due to repression of the *rot* repressor. (**c, d, e**) The lethal effects of immune challenge, which is called triggering (*Andrade-Linares et al., 2016*), are survived by the persistence ('memory') of the *agr*-activated state. (**f**) *agr* expression is inactivated by oxidation, thereby elevating expression of the antioxidant *bsaA* (*Sun et al., 2012*), which enables proliferation when oxidative stress is sublethal (*Sun et al., 2012*). (**g**) By surviving damage caused by lethal exogenous oxidative stress, primed *S. aureus* escape from the localized abscess to produce new infectious lesions (bloodstream dissemination) or to infect new hosts, where the cycle would be repeated.

ROS is generated during *agr* deficiency. Thus, protection from endogenous lethal stress via *agr* inactivation may not be only through the redox-dependent *bsaA* but also by a second pathway involving increased respiratory metabolism. The present work also supports the idea that exposing bacteria to exogenous $H_2O_2$ does not fully recapitulate the intracellular environment created by antibiotics and other stresses that act via ROS-mediated cell death (*Takahashi et al., 2017*), emphasizing that inactivation of *agr* can be either destructive or protective, depending on the type of lethal stress. Similar results have been reported with other bacteria: *mazF*, *lepA*, and *cpx* are destructive or protective based on the level of lethal stress (*Dorsey-Oresto et al., 2013*; *Wu et al., 2011*).

The protective activity of *agr* was carried over to in vivo studies using mice, as it was largely absent if the mice were deficient in host phagocyte production of ROS (*Cybb*$^{-/-}$ mice with a null allele of NADPH oxidase). The benefits of *agr* to *S. aureus* fitness seen with NADPH oxidase-proficient mice were observed largely in lungs, a key host defense niche for neutrophil-mediated clearance of disseminating pathogens (*Yipp et al., 2017*). The redox switch in AgrA, plus the protective properties associated with *agr* activation, lead to a clinical model in which *agr* links virulence-factor expression to an intrinsic protection against a lethal, $H_2O_2$-mediated immune response during infection (*Figure 11*). In this model, *agr* quorum-sensing renders cells better prepared to respond to lethal, exogenous oxidative stress. We note, however, that *agr*-mediated fitness benefits were present in certain tissues even in NADPH oxidase-deficient mice, indicating the existence of long-lived factors other than those that suppress oxidative stress. Thus, such a pre-emptive defense system may apply to many challenges experienced by *S. aureus* during infection, especially during bloodstream dissemination and conditions within inflamed tissues (*Richardson et al., 2008*; *Vitko et al., 2015*).

In conclusion, uncoupling of *agr*-mediated tolerance from bacterial population density anticipates increases in exogenous ROS expected during *S. aureus*-host interactions, thereby contributing to virulence. The ubiquity of quorum sensing suggests that it protects many bacterial species from oxidative damage. The next step is to find RNA, protein, and/or epigenetic markers underlying the *agr*-mediated 'memory' that improves protection against subsequent $H_2O_2$ exposure, since that will provide insights into the role of *agr* in cellular survival and adaptation during infection. Discovering ways to manipulate the lethal stress response, as seen with supplementation of antimicrobials with N-acetyl-cysteine during treatment of *Mycobacterium tuberculosis* (*Vilchèze and Jacobs, 2023*) and development of inhibitors of enzymes that produce protective $H_2S$ (*Shatalin et al., 2021*), could reveal novel approaches for enhancing antimicrobial therapy and host defense systems (*Cao et al., 2017*; *Gusarov et al., 2009*; *Shatalin et al., 2011*).

# Materials and methods

## Bacterial strains, plasmids, primers, and growth conditions

*S. aureus* strains, plasmids, and primers used in the study are described in *Tables 1 and 2*. Bacterial cells were grown in Tryptic Soy Broth (TSB, glucose concentration at 2.5 g/L) at 37 °C with rotary shaking at 180 rpm. For suboptimal aeration, broth cultures were grown in a closed-capped 15 mL conical tube with 10 mL of TSB. Colony formation was on Tryptic Soy Agar (TSA) with or without defibrinated sheep blood, incubated at 37 °C or 30 °C. Phages 80α and Φ11-mediated transduction was used for strain construction (*Novick, 1991*); transductants were selected on TSA plates containing the appropriate antimicrobial.

For analysis of in vitro growth curves, overnight cultures grown in TSB were diluted 1:1000 in CDM (*Hussain et al., 1991*), and growth was monitored at 37 °C in 96-well plates (100 µL/well) using an Agilent LogPhase 600 Microbiology Reader (Santa Clara, CA) with 1 mm orbital shaking, measuring $OD_{600}$ at 40 min intervals. The curves represent averaged values from five biological replicates. The exponential phase was used to determine growth rate (µ) from two datapoints, $OD_1$ and $OD_2$ flanking the linear portion of the growth curve, following the equation $lnOD_2-lnOD_1/t_2-t_1$, as described (*Grosser et al., 2016*).

## Measurement of bioluminescence

Overnight cultures were diluted to $OD_{600}$ ~0.05 and grown in TSB at 37 °C with rotary shaking at 180 rpm. Aliquots (100 µL) were inoculated into flat bottom 96-well microtiter plates (Corning,

**Table 1.** Bacterial strains[*].

| Strain | Background | Relevant genotype | Reference or source |
|---|---|---|---|
| BS819 | LAC | *agr* group I wild-type (CC8), Erm[S] | *Boles et al., 2010* |
| BS1348 | BS819 | *agr::tetM* | *Kumar et al., 2017* |
| BS820 | BS819 | *agr::ermC* | *Kumar et al., 2017* |
| BS821 | BS819 | *rnaIII::cadA* | *Wilde et al., 2015* |
| BS12 | Newman | *agr* group I wild-type (CC8) | *Duthie and Lorenz, 1952* |
| BS13 | BS12 | *agr::tetM* | *Geisinger et al., 2012* |
| BS669 | BS12 | *rnaIII::cadA* | *Kumar et al., 2017* |
| BS39 | BS39 clinical strain | *agr* (+) clinical isolate (CC45) | *Benson et al., 2011* |
| BS40 | BS40 clinical strain | *agr* (-) clinical isolate (CC45) | *Benson et al., 2011* |
| BS867 | JE2 | *agr* group I wild-type (CC8) | *Fey et al., 2013* |
| BS1010 | BS867 | *agr::cadA* in *S. aureus* JE2 | This study |
| BS1280 | BS12 | *saeQRS::spec* | *Benson et al., 2012* |
| BS1282 | BS12 | *agr::tetM, saeQRS::spec* | *Benson et al., 2012* |
| BS653 | *E. coli* | Top10 with pJC1111 (amp[R] in *E. coli*; Cd[R] in *S. aureus*) | *Chen et al., 2014* |
| BS656 | RN4220 | RN4220 with pRN7023 [shuttle vector (amp[R] in *E. coli*; Cm[R] in *S. aureus*) containing SaPI1 *int*] | *Chen et al., 2014* |
| BS435 | RN6734 | *agr* group-I prototype strain, derivative of NCTC 8325 | *Ji et al., 1997* |
| BS688 | BS435 | *agr::cadA* | This study |
| GAW130 | BS435 | *agr::cadA*, SaPI1 *attC::pGAW98 (agr-I ΔagrBD)* | This study |
| GAW183 | BS435 | *rnaIII::cad* | This study |
| BS450 | MW2 | *agr* group I wild-type (CC1) | *Baba et al., 2002* |
| BS451 | MW2 | *agr::tetM* | This study |
| BS988 | 126 a | *agr* (+) clinical isolate (CC5) | *Benson et al., 2011* |
| BS989 | 127b | *agr* (-) clinical isolate (CC5) | *Benson et al., 2011* |
| BS842 | BS819 | BS820 with SaPI1-*attC::agr*-IpJC1111 (*agr*-I, 8325–4) | *Kumar et al., 2017* |
| BS1301 | BS819 | *rot::Tn917* | This study |
| BS1302 | BS819 | *agr::tetM, rot::Tn917* | This study |
| BS1279 | BS12 | *rot::Tn917* | *Benson et al., 2012* |
| BS1281 | BS12 | *agr::tetM, rot::Tn917* | *Benson et al., 2012* |
| VJT14.28 | BS12 | *pOS1-Plgt-sodARBS-rot* | *Benson et al., 2012* |
| BS1486 | BS819 | *ahpC::bursa* (NE911) | This study, *Fey et al., 2013* |
| BS1487 | BS819 | *agr::tetM, ahpC::bursa* | This study, *Fey et al., 2013* |
| BS1488 | BS819 | *katA::bursa* (NE1366) | This study, *Fey et al., 2013* |
| BS1489 | BS819 | *agr::tetM, katA::bursa* | This study, *Fey et al., 2013* |
| BS1399 | BS12 | *sodA::tetM, sodM::ermC* | *Kehl-Fie et al., 2011* |
| BS1422 | BS819 | *sodA::tetM* | This study |
| BS1423 | BS819 | *agr::ermC, sodA::tetM* | This study |
| BS1435 | BS819 | *sigB* clean deletion | *Lauderdale et al., 2009* |
| BS1436 | BS819 | *agr::tetM, sigB* clean deletion | *Lauderdale et al., 2009* |

*Table 1 continued on next page*

*Table 1 continued*

| Strain | Background | Relevant genotype | Reference or source |
|---|---|---|---|
| BS1246 | BS12 | *mgrA::cat* | **Luong et al., 2006** |
| BS999 | BS819 | BS819 with SaPI1 *attC*::pGY*lux* (vector containing promoterless *lux*) | **Mesak et al., 2009** |
| BS1222 | BS819 | BS819 with SaPI1 *attC*::P*agrp3-lux* | **Figueroa et al., 2014** |
| BS1518 | BS12 | *agr::tetM, mgrA::cat* | This study |
| BS1527 | BS867 | *bshC::bursa* (NE230) | **Fey et al., 2013** |
| BS1528 | BS867 | *agr::tetM, bshC::bursa* | This study |
| BS1522 | BS867 | *gpxA2::bursa* (NE563) | **Fey et al., 2013** |
| BS1523 | BS867 | *agr::tetM, gpxA2::bursa* | This study |
| BS1490 | BS819 | *bsaA::bursa* (NE1730) | This study |
| BS1491 | BS819 | *bsaA::bursa, agr::tetM* | This study |
| BS1707 | BS819 | BS1422 with SaPI-*attC*::P*sarA-sodRBS-sodA* | This study |
| BS1708 | BS819 | BS1348 with SaPI-*attC*::P*sarA-sodRBS-sodA* | This study |
| BS1494 | BS867 | *ahpF::bursa* (NE1571) | **Fey et al., 2013** |
| BS1504 | BS867 | *agr::tetM, ahpF::bursa* | This study |
| BS1495 | BS867 | *SAUSA300_0377::bursa* (NE725) | **Fey et al., 2013** |
| BS1501 | BS867 | *agr::tetM, SAUSA300_0377::bursa* | This study |
| BS1496 | BS867 | *SAUSA300_0378::bursa* (NE537) | **Fey et al., 2013** |
| BS1502 | BS867 | *agr::tetM, SAUSA300_0378::bursa* | This study |
| BS1744 | BS819 | *acnA::bursa* (NE861) | This study |
| BS1745 | BS1348 | *agr::tetM, acnA::bursa* | This study |

*All bacterial strains are *S. aureus*, unless otherwise indicated. Abbreviations: CC, clonal complex; NEx, strain designation in the Nebraska Transposon Mutant Library (**Fey et al., 2013**).

Corning, NY), and bioluminescence was detected using a BioTek Synergy Neo2 plate reader (Agilent, Santa Clara, CA).

## Antimicrobials and chemicals

Antimicrobials, chemicals, and reagents were obtained from MilliporeSigma (Burlington, MA) or Thermo Fisher Scientific (Waltham, MA).

## Construction of mutants

Transposon mutants were generated by transducing *Bursa aurealis* insertions, obtained from the University of Nebraska transposon mutant (ΦNE) library (**Fey et al., 2013**), into LAC or LAC *agr::tetM* using phages 80α and Φ11.

Construction of the Δ*agrBD* mutant: *S. aureus* Δ*agrBD* mutant GAW128 was generated by a chromosomal integration strategy outlined in **Chen et al., 2014** in an *agr*-null background strain, BS687. Plasmid pJC1111, a suicide plasmid containing a cadmium resistance (Cd$^R$) cassette and the SaPI1 *attS* site that enables single-copy insertion into the corresponding chromosomal SaPI1 *attC* site, was used as the backbone vector for the *S. aureus agrBD* construct. pJC1000 contains the RN6734 *agr* locus cloned into pUC18. Inverse PCR of pJC1000 was performed using *agrBD* primers GWO#27 and GWO#28, re-ligated following treatment with polynucleotide kinase, and designated pGAW98. The *Sph*I-*EcoR*I fragment of pGAW98 was ligated into the SaPI1 integration vector pJC1111 and designated pGAW119. Strain RN9011 (RN4220 with pRN7023 [vector (Cm$^R$) containing SaPI1 integrase]) was electroporated with plasmid pGAW119 and plated on GL agar containing 0.1 mM CdCl$_2$. Phage

**Table 2.** Oligonucleotides.

| # | Name | Gene/Target | Sequence 5'→ 3' | Source |
|---|------|-------------|-----------------|--------|
| 1 | pflBRT.1a | | AAAAATGGAAGATGGAACAGACAC | *Kinkel et al., 2013* |
| 2 | pflBRT.1b | *pflB* | TCGATAACTGCATTACTTGTTCC | |
| 3 | pflART.1a | | TGACAAACATATTAGATTGACAGGAAAGC | *Chen et al., 2009* |
| 4 | pflART.1b | *pflA* | ATCATCAGAATAACCAGGCACAAGG | |
| 5 | ldh2RT.1a | | GGATCTGTAGGATCAAGCTATGCC | *Richardson et al., 2008* |
| 6 | ldh2RT.2b | *ldh2* | TGGTGAAGGACTGTGGACTGTACC | |
| 7 | nrdGRT.1a | | CAGTGTTTATGTATCAGGATGTCC | *Kinkel et al., 2013* |
| 8 | nrdGRT.1b | *nrdG* | GTTCGCCACCTAATAGACTTAGCC | |
| 9 | qoxB-RT.3A | | GTTGTACTTGGCATGTTCGCC | *Dmitriev et al., 2021* |
| 10 | qoxB-RT.3B | *qoxB* | GGCATTATGGTGCATCTTACC | |
| 11 | cydA-RT.1A | | CATTTCGATACATCTTCCCATGCC | *Dmitriev et al., 2021* |
| 12 | cydA-RT.1B | *cydA* | ATCTGCTAAGAAACTCAATAGTCC | |
| 13 | hmp-RT.1A | | TGACTTTAGTGAATTTACACCAGG | *Dmitriev et al., 2021* |
| 14 | hmp-RT.1B | *hmp* | CGTTTAACGCCAAAAGTTAAATGG | |
| 15 | spaRT1 | | CAAACCTGGTCAAGAACTTGTTGTTG | *Brignoli et al., 2019* |
| 16 | spaRT2 | *spA* | GCTAATGATAATCCACCAAATACAGTTG | |
| 17 | clfB RT1 | | GGATAGGCAATCATCAAGCACAAG | *Brignoli et al., 2019* |
| 18 | clfB RT2 | *clfA* | GCTATCTACATTCGCACTGTTTGTG | |
| 19 | ahp RT For | | CGTAAAAACCCTGGCGAAGTAT | *Mashruwala and Boyd, 2017* |
| 20 | ahp RT Rev | *ahpC* | TGCAATGTTTTAGCGCCTTCT | |
| 21 | kat RT For | | TGGTGTTTTTGGGCATCCA | *Shee et al., 2022* |
| 22 | kat RT Rev | *katA* | CCCTAGGCCCTGCTGTCATA | |
| 23 | rpoB F | | GAACATGCAACGTCAAGCAG | *Dyzenhaus et al., 2023* |
| 24 | rpoB R | *rpoB* | AATAGCCGCACCAGAATCAC | |
| 25 | MPsodA#1 | | AGGCGCGCCTTTATTTTGTTGCAT TATATAATTCGTCAACTTTTTCCCAG | This study |
| 26 | MPsodA#2 | *sodA* | GGATGATTATTTATGGCTTTTGAATT ACCAAAATTACCATACGC | This study |
| 27 | MPsodA#3 | | TTCAAAAGCCATAAATAATCATCCTCCT AAGGTACCCGG | This study |
| 28 | MPsodA#4 | *PsarA* | GCGGCCGCTCTGATATTTTTGACTA AACCAAATGCTAACCCAG | This study |
| 29 | MPsodA#5 | | AAAATATCAGAGCGGCCGCCAG | This study |
| 30 | MPsodA#6 | pJC1111 | ACAAAATAAAGGCGCGCCTATTCTAAATG | This study |
| 31 | pJC1111 FOR | pJC1111-P*sarA*-sodRBS-sodA | TGGCCTTTTGCTCACATGTTCTTT CCTGCGTTATCCCCTGATTC | This study |
| 32 | pJC1111 REV | | TGATATCAAAATTATACATGTCAACG | This study |
| 33 | GWO#27 | | CAATTTTACACCACTCTCCTC ACTGTCATTATACGATTTAG | This study |
| 34 | GWO#28 | *agrBD* | TAATTTAAATAGAGAGTGTGAT AGTAGGTGGAATTATTAAATAG | This study |

*Table 2 continued*

| # | Name | Gene/Target | Sequence 5'→ 3' | Source |
|---|------|-------------|-----------------|--------|
| 35 | JCO#339 | | GGTACCTGAAGCGGGCGAGCGAG | This study |
| 36 | JCO#340 | | GGATCCGATAATAAAGTCAGTTAACGACG TATTCAATTGTAAATCTTGTTGG | This study |
| 37 | JCO#342 | | CTCGAGAAGAAGGGATGAGTTAATCATCATTATGAGAC | This study |
| 38 | JCO#343 | *agr* flanking regions | GCATGCGATCTATCAAGGATGTGATGTTA TGAAAGTCCAAATTTATCAATTACCG | This study |

80α lysates of Cd^R colonies were used to transduce BS687 (RN6734 Δ*agr::ermC*, Erm), generating GAW128 (Δ*agrBD*).

To construct *agr* mutant BS687, *agr* flanking regions were amplified with primer pairs JCO#339, JCO#340, and JCO#342, JCO#343 and cloned into the *Hinc*II site of pUC18 to generate pJC1527 and pJC1528, respectively. pJC1530 was generated by four-way ligation of the *Kpn*I-*Bam*HI fragment of pJC1527 (*agr* left flank), *Xho*I-*Sph*I fragment of pJC1528 (*agr* right flank), and *Bam*HI-*Xho*I fragment from pJC1073 (Erm cassette) to *Kpn*I-*Sph*I digested pJC1202 (replacement vector). Plasmid pJC1530 was electroporated into strain RN4220 with selection on GL agar containing 10 µg/mL of chloramphenicol at 30 °C. Phage 80α lysates of Cm^R colonies were used to transduce strain JCSA18 (*rpsL*\* mutant of RN6734 that results in streptomycin resistance) and then allelic exchange of the Em^R Sm^S Cm^R colonies was performed as previously described. Phage 80 a lysates of Em^R Sm^R Cm^S colonies were then used to transduce RN6734 with selection for Em^R, generating BS687. *sodA* complementation: Plasmid P*sarA*-*sodA*-pJC1111, expressing *sodA* under the control of the constitutive promoter P*sarA*, was integrated into the *S. aureus* chromosome at the SaPI1 *attC* site of strain LAC (*Geisinger et al., 2008*), LAC *sodA::tetM*, and LAC *agr::ermC*. Complementation plasmid P*sarA*-*sod*RBS-*sodA* was generated by Gibson assembly and inserted into the SaPI1 integration vector pJC1111. Wild-type *sodA* and the *sarA* promoter were amplified from *S. aureus* gDNA using primers MPsodA#1–2 (*sodA* gene and RBS) and MPsodA#3–4 (P*sarA*). Primers MPsodA#5–6 were used to linearize pJC1111. Primers introduced relevant oligonucleotide overlaps that enabled Gibson assembly (*Shee et al., 2022*), generating P*sarA*-*sod*RBS-*sodA*. P*sarA*-*sod*RBS-*sodA* was transformed into *E. coli* DH5α for amplification, purification, and sequence validation via primers pJC1111 FOR and pJC1111 REV. Purified P*sarA*-*sod*RBS-*sodA* was electroporated into RN9011 and positive chromosomal integrants at the SaPI1 chromosomal attachment (*attC*) site were selected with 0.1 mM CdCl$_2$. Phage 80 a lysates of positive integrants were used to transduce BS1422 (LAC *sod::tetM*) and BS1348 (LAC *agr::tetM*), generating BS1707 and BS1708, respectively.

## Survival measurements

To measure lethal action, overnight cultures were diluted (OD$_{600}$~0.05) in fresh medium and grown with shaking to early exponential (OD$_{600}$~0.15) or late log (OD$_{600}$~4) phase, conditions when *agr* expression is largely absent (*Kumar et al., 2017*) or maximally activated, respectively. Early (undiluted) and late exponential phase cultures (diluted into fresh TSB medium to OD$_{600}$~0.15) were incubated with H$_2$O$_2$ under aerobic conditions either at a fixed concentration for one or more time points or at various concentrations for a fixed time. At the end of treatment, aliquots were removed, concentrated by centrifugation and serially diluted in phosphate-buffered saline to remove H$_2$O$_2$, and plated for determination of viable counts at 24 hr. The percentage of survival was calculated relative to a sample taken at the time of H$_2$O$_2$ addition. When menadione and N-acetylcysteine were used to inhibit or potentiate killing by H$_2$O$_2$, they were added prior to lethal treatments as described previously (*Conlon et al., 2016*). For experiments involving menadione pretreatment, cultures were grown for 3.5 hr, and menadione (40 mM solution in 96% EtOH, final concentration 80 µM) was added for the last 0.5 hr of culture, preceding the H$_2$O$_2$ treatment at 4 hr. N-acetylcysteine was used to counter the action of menadione; it was added simultaneously with menadione, at a final concentration of 30 mM (640 mM stock in sterile ddH$_2$O was used). All experiments were repeated at least three times; similar results were obtained from the biological replicates.

### Measurement of glucose consumption

Overnight cultures were diluted into fresh TSB ($OD_{600}$ 0.05) and grown for 4 hr with shaking at 180 rpm ($OD_{600}$ 4) at 37 °C. Glucose was assayed in the supernatant fluids of bacterial cultures following centrifugation at 12,000 × g, using Centricon-10 concentrators (MilliporeSigma, Burlington, MA), and pH adjustment to 6.5–7.0 using NaOH. Cells were assayed and plated hourly for determination of viable counts as indicated in figures. Glucose content was measured from serial dilutions of supernatants using the UV method (cat. no. 10-716-251-035) following manufacturer's instructions (R-Biopharm, Darmstadt, Germany). Glucose consumption was expressed as μg of glucose consumed over 3 hr of culture per $10^8$ bacterial cells. There was no detectable glucose in culture supernatants at 4 hr of culture (data not shown).

### Measurement of excreted metabolites

Excreted metabolites were assayed in the supernatant fluids of bacterial cultures following centrifugation at 12,000 × g for 10 min for late exponential (4 hr, $OD_{600}$~4) or multiple time points (acetate), as indicated in figures. Aliquoted supernatants were stored at −80 °C and thawed on ice prior to analysis. Cells were plated for determination of viable counts; L(+)-lactate and acetate concentrations were measured using commercially available colorimetric and fluorometric kits (cat. no. MAK065, ab204719), according to manufacturer's recommendations (MilliporeSigma, Burlington, MA and Abcam, Cambridge, UK, respectively).

### Measurement of intracellular metabolites

Overnight cultures were diluted into fresh TSB ($OD_{600}$~0.05), grown for 4 hr at 37 °C with shaking at 180 rpm ($OD_{600}$~4), and plated for determination of viable counts at 4 hr. The remaining cells were concentrated by centrifugation at 12,000 x× g for 10 min, and resuspended in lysis buffer provided by the assay kit. Cells were lysed by repeated homogenization (two cycles of 45 s homogenization time at 6 M/s followed by a 5 min pause on ice) using Lysing Matrix B tubes in a FastPrep-24 homogenizer (MP Biomedicals, Irvine, CA). After lysis, cell debris was removed by centrifugation (12,000 × g, 10 min) and the supernatant was used for determination of pyruvate, fumarate, citrate, and acetyl-CoA levels using colorimetric (pyruvate, fumarate), or flurometric (citrate, acetyl-CoA) assays (cat. no. KA1674, ab102516, KA3791, and MAK039, respectively) and a microplate reader (BioTek Synergy Neo2, Agilent, Santa Clara, CA) according to the manufacturer's instructions (Abnova, Taipei City, Taiwan and Abcam, Cambridge, UK and MilliporeSigma, Burlington, MA, respectively). Assayed metabolites were measured in μg and normalized to cell count.

### Measurement of oxygen consumption

Overnight cultures were diluted into fresh TSB ($OD_{600}$~0.05), grown for 5 hr at 37 °C with shaking at 180 rpm ($OD_{600}$~4), diluted ($OD_{600}$~0.025) in fresh TSB, and added to a microtiter plate (200 μL/well). Oxygen consumption rate (OCR) was measured using a Seahorse XF HS Mini Analyzer (Agilent, Santa Clara, CA) according to the manufacturer's instructions. The Seahorse XF sensor cartridge was hydrated in a non-$CO_2$ 37 °C incubator with sterile water (overnight) and pre-warmed XF calibrant for 1 hr prior to measurement. OCR measurements were recorded in 15 measurement cycles with 3 min of measurement and 3 min of mixing per cycle. CFU were enumerated to confirm equal concentrations of *agr*-deficient mutant and wild-type cells.

### Measurement of ATP, NAD+, and NADH

For ATP, overnight cultures were diluted into fresh TSB ($OD_{600}$~0.05), grown for 4 hr at 37 °C with shaking at 180 rpm ($OD_{600}$~4), diluted ($OD_{600}$~1.0) in fresh TSB, and incubated at room temperature with reagent for determination of ATP using BacTiter-Glo Microbial Cell Viability Assay (cat. no. G8232; Promega, Madison, WI), according to the manufacturer's instructions. Luminescence was detected in a BioTek Synergy Neo2 plate reader (Agilent, Santa Clara, CA). The amount of ATP was calculated and normalized to cell count.

For NAD+ and NADH, overnight cultures were diluted into fresh TSB ($OD_{600}$~0.05), grown for 4 hr at 37 °C with shaking at 180 rpm ($OD_{600}$~4), and plated for viable counts at 4 hr or concentrated by centrifugation at 12,000 × g for 10 min and resuspended in lysis buffer provided by the assay kit. Cells were lysed by repeated homogenization (two cycles of 45 s homogenization time at 6 M/s followed

by a 5 min pause on ice) using Lysing Matrix B tubes in a FastPrep-24 homogenizer (MP Biomedicals, Irvine, CA). After lysis, cell debris was removed by centrifugation ($12,000 \times g$, 10 min), and the supernatant was used for determination of $NAD^+$ and NADH levels using a colorimetric assay kit (cat. no. KA1657; Abnova, Taipei City, Taiwan) and a microplate reader (BioTek Synergy Neo2, Agilent, Santa Clara, CA) according to the manufacturer's instructions.

## Measurement of baseline ROS levels

Overnight cultures were diluted into fresh TSB (OD$_{600}$~0.05), and grown with shaking to early exponential phase (OD$_{600}$~0.2). 200 µL of culture was removed and cell density was normalized before staining with carboxy-H2DCFDA fluorescent dye (final concentration 10 µM) (Invitrogen, Waltham, MA). Samples were incubated at room temperature for 5 min, then 800 µL of PBS + EDTA buffer (100 mM) was added to each sample, and ROS levels were measured by fluorescence-based flow cytometry (BD Fortessa, BD Biosciences, San Jose, CA). All tubes with cultures were wrapped with aluminum foil to avoid light. A sample containing LAC wild-type cells lacking carboxy-H2DCFDA was included as a control for auto-fluorescence. Forward and side scatter parameters were acquired with logarithmic amplification. ROS was detected using the 488 laser and a 530/30 nm bandpass filter. Data were analyzed using FlowJo software version 10.8.1 (BD Biosciences, San Jose, CA).

## Measurement of superoxide dismutase (SOD) activity

Overnight cultures were diluted (OD$_{600}$~0.05) into fresh TSB, grown to late exponential phase (OD$_{600}$~4), diluted to OD$_{600}$=1, centrifuged at $12,000 \times g$ for 5 min, and the cell pellet was homogenized in 300 µL of ice-cold lysis buffer (100 mM Tris-HCl pH 7.4 + 0.5% Triton + 5 mM 2-mercaptoethanol + 0.2 mM PMSF). SOD activity was measured using a commercially available kit (cat. no CS0009-1KT), according to the manufacturers' instructions (MilliporeSigma, Burlington, MA). The experiment was repeated three times with similar results.

## RNA sequencing and data analysis

Overnight cultures were diluted (OD$_{600}$~0.05) into fresh TSB medium and grown at 37 °C to early exponential phase (OD$_{600}$~0.5) (*Δagr* single mutant and *Δagr Δrot* double mutant) or late exponential phase (OD$_{600}$~4) (wild-type and *Δagr* strains). Samples of *Δagr* and *Δagr Δrot* were divided into two 3 mL aliquots, and the aliquots were incubated at 37 °C for another 30 min, with or without treatment with H$_2$O$_2$. Peroxide concentrations for *Δagr* and *Δagr Δrot* were normalized to expected killing at the time of harvest (*Figure 7—figure supplement 1*).

Three independent cultures for each sample were used for determination of transcriptional profiles. Briefly, cultures were concentrated by centrifugation ($12,000 \times g$ for 5 min), and resuspended cells were disrupted using Lysing Matrix B tubes in a FastPrep-24 homogenizer (MP Biomedicals, Irvine, CA) at 6 M/s, for 30 s, three times (samples were resting on ice between homogenizer runs), and RNAs were extracted from the collected bacterial cells using TRIzol reagent (Thermo Fisher Scientific, Waltham, MA). RNA was isolated using RNeasy (Qiagen, Germantown, MD) mini spin columns. Sequence libraries were generated using the TruSeq Stranded Total RNA Library Prep kit (Illumina, San Diego, CA) following the manufacturer's recommendations. The rRNAs were removed by the Ribo-zero Kit (Illumina) to enrich mRNA, using 13 cycles of PCR amplification of the final library. Amplified libraries were purified using AMPure beads (Beckman Coulter, Brea, CA), quantified by Qubit (Thermo Fisher Scientific, Waltham, MA) and qPCR, and visualized in an Agilent Bioanalyzer (Santa Clara, CA). Pooled libraries were sequenced as paired-end 50 bp reads using an Illumina NovaSeq instrument.

Reads were initially trimmed using Trimmomatic version 0.39 (*Bolger et al., 2014*) to remove adaptors as well as leading or trailing bases with a quality score less than 3, filtering reads with a minimum length of 36. Reads were mapped to reference strain USA300 FPR3757 (RefSeq identifier GCF_000013465.1) using Bowtie2 version 2.2.5 (*Langmead and Salzberg, 2012*). Using gene annotations from the same assembly, reads mapped to each gene were counted with featureCounts version 2.0.1 (*Liao et al., 2014*), producing a counts matrix. Additional analysis was performed in R (R Core Team 2021) using the package DESeq2 version 1.32 (*Love et al., 2014*).

Normalization to account for inter-sample library size variation was performed using the built-in normalization function of DESeq2. All RNA-seq heatmaps were colored according to row (gene) z-scores of DESeq2 normalized counts. For differential expression testing, the Wald test was used with

a log2 fold-change threshold of 0.5 and an FDR of 0.1. For simple pairwise comparisons (e.g. the effect of strain under control conditions), datasets were split so that analysis was performed independently for strains used in the comparison. To determine the interaction between strain and condition variables, all samples were included with the experimental design (formula 'expression ~condition + strain + condition:strain,' where condition:strain is the interaction between variables).

### Metabolic flux prediction

The SPOT (Simplified Pearson cOrrelation with Transcriptomic data) computational method (*Kim et al., 2016*) was used to analyze the difference in intracellular metabolic fluxes between wild-type LAC and *agr::tetM* mutant grown in TSB to late exponential phase ($OD_{600} \sim 4$). SPOT is similar to the E-Flux2 method described previously (*Balasubramanian et al., 2016*), but a recent validation study (*Bhadra-Lobo et al., 2020*) shows that SPOT generally outperforms E-Flux2. SPOT infers metabolic flux distribution by integrating transcriptomic data in a genome-scale metabolic model of *S. aureus* (*Becker and Palsson, 2005*) that was adapted for use with strain LAC. For a list of the metabolic reactions ranked by unit of flux per 100 units of glucose uptake flux, see *Supplementary file 3*.

### Real-time qRT-PCR assays

Briefly, RNA was purified as described above from late exponential ($OD_{600} \sim 4.0$) cells, cDNAs were synthesized using Maxima First Strand cDNA Synthesis Kit (Thermo Fisher Scientific, Waltham, MA), and real-time reverse transcription quantitative PCR (qRT-PCR) was performed using QuantiNova SYBR Green PCR Kit (Qiagen, Hilden, Germany). Primers were synthesized by IDT Inc (Coralville, IA). Three independent biological samples were run in triplicate and *rpoB* was used to normalize gene expression. $2^{-\Delta\Delta Ct}$ method was used to calculate the relative fold gene expression (*Livak and Schmittgen, 2001*).

### Peritoneal infection of mice

C57BL/6 mice and C57BL/6 *Cybb*[-/-] (also known as gp91phox/nox2) were purchased from the Jackson Laboratory and bred onsite to generate animals for experimentation. Age and gender-matched, 8–10 week-old mice were used. *S. aureus* strains harboring *RNAIII* or *agrBD* deletion in the NCTC 8325 background were grown overnight in TSB (37 °C, 180 rpm) separately or mixed at a 1:1 ratio. Overnight cultures were diluted ($OD_{600} \sim 0.05$) into fresh TSB medium (subcultured separately for the cultures mixed overnight ('primed') or mixed 1:1 for *RNAIII* or *agrBD* mutant single cultures ('unprimed') and grown at 37 °C to early exponential phase ($OD_{600} \sim 0.5$)). Bacteria were washed one time by centrifugation with PBS and adjusted to $10^9$ CFU/mL. Twenty C57BL/6 WT mice and 17 *Cybb*[-/-] mice were injected intraperitoneally with 100 µL of either 'primed' or 'unprimed' inoculum. After 2 hr, internal organs, peritoneal lavage, and blood were collected. The organs were homogenized in sterile PBS and serial dilutions were plated for viable counts on TS agar. Collected blood was lysed with saponin and plated for viable counts on TSA plates. Peritoneal lavage fluid was serially diluted and plated for viable counts. All animal studies were performed as per an NYU Grossman School of Medicine Institutional Animal Care and Use Committee (IACUC) approved protocol for the Shopsin Lab.

### Statistical analysis

Prism software (GraphPad, Inc) was used to perform statistical analyses.

Statistical significance was determined using the Student's *t*-test, Mann–Whitney *U* test, one-way analysis of variance (ANOVA), or the Kruskal-Wallis test, depending on the data type. Statistical significance was considered to be represented by p values of <0.05.

## Acknowledgements

We thank Andrew Darwin for his critical comments on the manuscript. This work was supported by NIH National Institute of Allergy and Infectious Diseases grants R01AI137336 (BS, IY, and VJT); R01AI140754 (BS and VJT); R01AI150893 and R01AI038446 (JNW); R01AI149350 (VJT); K08AI163457 (RJU), T32AR064184 (TKK), and R21AI153646 (DP); New Jersey Health Foundation PC 142–22 and New Jersey Commission on Cancer Research COCR22RBG005 grants (DP); and funds from the NYU

Langone Health Antimicrobial-Resistant Pathogens Program (BS, AP, and VJT). The NYU Langone Health Genome Technology Center, and the Cytometry and Cell Sorting Laboratory are shared resources that are partially supported by the Cancer Center Support Grant P30CA016087 at the Laura and Isaac Perlmutter Cancer Center.

## Additional information

### Competing interests

Ashley L DuMont: Inventor on patents and patent applications (US8431, 687B2; US2019135900 A1; EP4313303A1) filed by New York University, which are currently under commercial license to Janssen Biotech Inc. Janssen Biotech Inc provides research funding and other payments associated with a licensing agreement. These patents pertain solely to the development of vaccines and therapeutics targeting S. aureus toxins and are unrelated to the content presented in this work. Victor J Torres: Has received honoraria from Pfizer and MedImmune, and is an inventor on patents and patent applications filed by New York University,(US8431, 687B2; US2019135900 A1; EP4313303A1) which are currently under commercial license to Janssen Biotech Inc. Janssen Biotech Inc provides research funding and other payments associated with a licensing agreement. Bo Shopsin: Has consulted for Basilea Pharmaceutica. The other authors declare that no competing interests exist.

### Funding

| Funder | Grant reference number | Author |
|---|---|---|
| National Institute of Allergy and Infectious Diseases | R01AI137336 | Itai Yanai<br>Victor J Torres<br>Bo Shopsin |
| National Institute of Allergy and Infectious Diseases | R01AI140754 | Victor J Torres<br>Bo Shopsin |
| National Institute of Allergy and Infectious Diseases | R01AI150893 | Jeffrey N Weiser |
| National Institute of Allergy and Infectious Diseases | R01AI038446 | Jeffrey N Weiser |
| National Institute of Allergy and Infectious Diseases | R01AI149350 | Victor J Torres |
| National Institute of Allergy and Infectious Diseases | K08AI163457 | Robert J Ulrich |
| National Institute of Allergy and Infectious Diseases | T32AR064184 | Theodora K Karagounis |
| National Institute of Allergy and Infectious Diseases | R21AI153646 | Dane Parker |
| New Jersey Health Foundation | PC 142-22 | Dane Parker |
| New Jersey Commission on Cancer Research | COCR22RBG005 | Dane Parker |
| NYU Langone Health Antimicrobial-Resistant Pathogens Program | | Alejandro Pironti |

The funders had no role in study design, data collection and interpretation, or the decision to submit the work for publication.

### Author contributions

Magdalena Podkowik, Conceptualization, Data curation, Formal analysis, Investigation, Visualization, Methodology, Writing – original draft, Writing – review and editing; Andrew I Perault, Erin E Zwack, Robert J Ulrich, Theodora K Karagounis, Chunyi Zhou, Julia Shenderovich, Junbeom Kwon, Investigation, Writing – review and editing; Gregory Putzel, Data curation, Formal analysis, Validation,

Methodology, Writing – review and editing; Andrew Pountain, Data curation, Formal analysis, Methodology, Writing – review and editing; Jisun Kim, Investigation, Visualization, Writing – review and editing; Ashley L DuMont, Investigation, Methodology, Writing – review and editing; Andreas F Haag, Xilin Zhao, Conceptualization, Writing – review and editing; Gregory A Wasserman, Jeffrey N Weiser, Resources, Writing – review and editing; John Chen, Resources, Investigation, Writing – review and editing; Anthony R Richardson, Conceptualization, Investigation, Writing – review and editing; Carla R Nowosad, Formal analysis, Investigation, Visualization, Writing – review and editing; Desmond S Lun, Formal analysis, Visualization, Writing – review and editing; Dane Parker, Alejandro Pironti, Supervision, Writing – review and editing; Karl Drlica, Conceptualization, Writing – original draft, Writing – review and editing; Itai Yanai, Supervision, Funding acquisition; Victor J Torres, Resources, Supervision, Funding acquisition; Bo Shopsin, Conceptualization, Resources, Supervision, Funding acquisition, Writing – original draft, Project administration, Writing – review and editing

## Author ORCIDs
Magdalena Podkowik ⓘ http://orcid.org/0009-0005-6772-3697
Andrew Pountain ⓘ http://orcid.org/0000-0001-9651-5145
Theodora K Karagounis ⓘ https://orcid.org/0000-0003-1481-9589
Itai Yanai ⓘ http://orcid.org/0000-0002-8438-2741
Victor J Torres ⓘ https://orcid.org/0000-0002-7126-0489
Bo Shopsin ⓘ https://orcid.org/0009-0001-7729-8584

## Ethics

This study was performed in strict accordance with the recommendations in the Guide for the Care and Use of Laboratory Animals of the National Institutes of Health. All of the animals were handled according to approved institutional animal care and use committee (IACUC) protocol (IA16-01941) of NYU Langone Health. The protocol was approved by the The Animal Care and Use Program at the NYU Grossman School of Medicine (Assurance number: D16-00274). Every effort was made to minimize suffering.

Reviewer #1 (Public review): https://doi.org/10.7554/eLife.89098.4.sa1
Reviewer #2 (Public review): https://doi.org/10.7554/eLife.89098.4.sa2
Author response https://doi.org/10.7554/eLife.89098.4.sa3

---

# Additional files

## Supplementary files

• Supplementary file 1. RNA-seq comparison of agr wild-type and Δagr mutant strains grown to late exponential phase.

• Supplementary file 2. Data used for metabolic flux prediction.

• Supplementary file 3. RNA-seq comparison of ΔagrΔrot and Δagr mutant strains grown to early exponential phase, with or without treatment with $H_2O_2$.

• MDAR checklist

## Data availability

Sequencing data have been deposited in GEO under accession code GSE207045.

The following dataset was generated:

| Author(s) | Year | Dataset title | Dataset URL | Database and Identifier |
|---|---|---|---|---|
| Podkowik M, Perault A, Putzel G, Pountain A, Kim J, DuMont A, Zwack E, Ulrich R, Ulrich R, Karagounis T, Zhou C, Haag A, Shenderovich J, Wasserman G, Kwon J, Chen J, Richardson AR, Weiser J, Nowosad C, Lun D, Parker D, Pironti A, Zhao X, Drlica K, Yanai I, Torres VJ | 2023 | The quorum-sensing agr system protects *Staphylococcus aureus* from oxidative stress | https://www.ncbi.nlm.nih.gov/geo/query/acc.cgi?acc=GSE207045 | NCBI Gene Expression Omnibus, GSE207045 |

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
