## [Editor Report · eLife assessment]

This **important** study outlines how the agr quorum sensing system in *Staphylococcus aureus* confers long-lived protection against oxidative stress, thereby linking bacterial metabolism to virulence in this pathogen. While the findings, which are supported by **solid** data, seem at first glance to contradict earlier findings that show increased fitness of agr mutants under oxidative stress, the core conclusions of the study are well-substantiated. The topic of the paper holds broad relevance to microbiologists, especially those focusing on host-pathogen interactions and bacterial responses to ROS.

---

## [Referee Report · Reviewer #1 (Public review)]

As a pathogen, *S. aureus* has evolved strategies to evade the host's immune system. It effectively remains 'under the radar' in the host until it reaches high population densities, at which point it triggers virulence mechanisms, enabling it to spread within the host. The agr quorum sensing system is central to this process, as it coordinates the pathogen's virulence in response to its cell density.

In this study, Podkowik and colleagues suggest that cells activating agr signaling also benefit from protection against H2O2 stress, whereas inactivation of agr increases cell death. The underlying cause of this lack of protection is tied to an ATP deficit in the agr mutant, leading to increased glucose consumption and NADH production, ultimately resulting in a redox imbalance. In response to this imbalance, the agr mutant increases respiration, resulting in the endogenous production of ROS which synergizes with H2O2 to mediate killing of the agr mutant. Suppressing respiration in the agr mutant restored protection against H2O2 stress.

Additionally, the authors establish that agr-dependent protection against oxidative stress is also linked to RNAIII activation, and the subsequent block of Rot translation. However, the specific protective genes regulated by Rot remain unidentified. Thus, according to the evidence provided, agr triggers intrinsic mechanisms that not only decrease harmful ROS production within the cell but also alleviate its detrimental effects.

Interestingly, these protective mechanisms are long-lived, and guard the cells against external oxidative stressors such as H2O2, even after the agr system has been 'turned off' in the population.

---

## [Referee Report · Reviewer #2 (Public review)]

In their study, Podkowik et al. elucidate the protective role of the accessory gene regulator (agr) system in *Staphylococcus aureus* against hydrogen peroxide (H2O2) stress. Their findings demonstrate that agr safeguards the bacterium by controlling the accumulation of reactive oxygen species (ROS), independent of agr activation kinetics. This protection is facilitated through a regulatory interaction between RNAIII and Rot, impacting virulence factor production and metabolism, thereby influencing ROS levels. Notably, the study highlights the remarkable adaptive capabilities of *S. aureus* conferred by agr. The protective effects of agr extend beyond the peak of agr transcription at high cell density, persisting even during the early log-phase. This indicates the significance of agr-mediated protection throughout the infection process. The absence of agr has profound consequences, as observed by the upregulation of respiration and fermentation genes, leading to increased ROS generation and subsequent cellular demise. Interestingly, the study also reveals divergent effects of agr deficiency on susceptibility to hydrogen peroxide compared to ciprofloxacin. While agr deficiency heightens vulnerability to H2O2, it also upregulates the expression of bsaA, countering the endogenous ROS induced by ciprofloxacin. These findings underscore the complex and context-dependent nature of agr-mediated protection. Furthermore, in vivo investigations using murine models provide valuable insights into the importance of agr in promoting *S. aureus* fitness, particularly in the context of neutrophil-mediated clearance, with notable emphasis on the pulmonary milieu. Overall, this study significantly advances our understanding of agr-mediated protection in *S. aureus* and sheds light on the sophisticated adaptive mechanisms employed by the bacterium to fortify itself against oxidative stress encountered during infection.

The conclusions drawn in this paper are generally well-supported by the data.

---

## [Author Response]

The following is the authors’ response to the previous reviews.

**Reviewer #1 (recommendations for the authors):**
Additional suggestions for improvement are noted below:(1) Additional 1. Lns 261-262, as well as abstract: The term 'aerobic fermentation' is not accurate in the context of this manuscript. This terminology should be reserved for conditions where lactate production is observed under optimal aerobic conditions. This is not the case in this study. More lactate was observed in the agr mutant only when cells were grown under microaerobic conditions, where some level of fermentation would be expected to be active (esp. if nitrate is not provided in media).

We modified the text by deleting reference to the “aerobic” fermentation as suggested by the reviewer:

Line 93 (abstract): “Deletion of agr increased both respiration and aerobic fermentation but decreased ATP levels and growth, suggesting that Δagr cells assume a hyperactive metabolic state in response to reduced metabolic efficiency.”

Line 184: “Collectively, these data suggest that Δagr increases respiration and aerobic fermentation to compensate for low metabolic efficiency.”

(2) Additionally, the authors' statement, 'The tendency of Δagr cells to forgo the additional ATP yield from acetate production in favor of NAD+-generating lactate (23, 24) underscores the importance of redox balance in Δagr cells,' appears contradictory to the data presented in Fig 5, where the Δagr mutant demonstrates an approximately threefold increase in acetate production during exponential growth compared to the wild-type strain. A clarification or adjustment in the manuscript may be necessary to ensure consistency and accurate interpretation.

In glucose-fermenting *S. aureus*, pyruvate can serve as an electron acceptor, generating lactate from lactate dehydrogenases. Acetyl-CoA production proceeds via the pyruvate formate-lyase reaction, which converts pyruvate to formate rather than CO2 and thus does not consume oxidized NAD+. Thus, at a general level, the tendency of fermenting cells to forgo the additional ATP yield from acetate production in favor of NAD+-generating ethanol synthesis underscores the importance of redox balance when respiration is suboptimal. This is especially true for fermenting Δagr strains, as evidenced by increased lactate production compared to their relatively ATP replete wild-type parental strains. However, in the interest of clarity, we removed the sentence in question, because it is not necessary and potentially confusing, and because the additional context it requires would detract from the manuscript by disrupting its sense of narrative and brevity.

(3) Ln 277-285: There still are errors in how this paragraph is worded. What the authors stated in the 'response to the reviewers' (question 13) and the changes they made in the text are different. Here again, the response to question 13 suggested the following, "Collectively, these observations suggest that a surge in NADH production and reductive stress in the Δagr strain induces a burst in respiration, but levels of NADH are saturating, thereby driving fermentation in the presence of oxygen." That bit of it where the authors suggest that fermentation was activated because NADH was saturating is only true under microaerobic conditions and not under oxygen rich conditions.Reviewer #1 (comment under Review): Data presented in Figure 5 suggest the opposite - a surge in NADH accumulation leading to a decrease in the NAD/NADH ratio, rather than a surge in the 'consumption' of NADH. Clarifying this point in the manuscript would ensure accurate representation of the findings.

Responses to Comments 3 and a comment in the Review have been combined.

Line 280: We thank the Reviewer for their attention to detail in picking up our error in response to question 13 related to the difference in the revised text and “response to reviewers”. We modified the text accordingly.

“Microaerobic conditions and “consumption”: We have modified the wording and fixed the error with respect to “consumption” as pointed out by the reviewer (strikethrough/underlined):

Line 285: “Collectively, these observations suggest that a surge in NADH consumption accumulation and reductive stress in the Δagr strain induces a burst in respiration, but levels of NADH are saturating, thereby driving fermentation under microaerobic conditions in the presence of oxygen.”

**Reviewer #2 (recommendations for the authors):**
(1) The authors are requested to revise 'we expected a lower NAD+/NADH' in line 280 to 'we expected a higher NAD+/NADH.' Additionally, what was the glucose concentration in TSB media?

NAD+/NADH: We thank the Reviewer for their attention to detail in picking up our error. Our responses to Reviewer 1, Comment 3 above addresses this issue.

Glucose: We modified the Methods as suggested.